# Fluids as primary carriers of sulphur and copper in magmatic assimilation

Ville J. Virtanen [1✉], Jussi S. Heinonen [1], Ferenc Molnár[2,8], Max W. Schmidt[3], Felix Marxer [3,4], Pietari Skyttä[5], Nico Kueter[6] & Karina Moslova[7]

Magmas readily react with their wall-rocks forming metamorphic contact aureoles. Sulphur and possibly metal mobilization within these contact aureoles is essential in the formation of economic magmatic sulphide deposits. We performed heating and partial melting experiments on a black shale sample from the Paleoproterozoic Virginia Formation, which is the main source of sulphur for the world-class Cu-Ni sulphide deposits of the 1.1 Ga Duluth Complex, Minnesota. These experiments show that an autochthonous devolatilization fluid effectively mobilizes carbon, sulphur, and copper in the black shale within subsolidus conditions ($\leq 700\ °C$). Further mobilization occurs when the black shale melts and droplets of Cu-rich sulphide melt and pyrrhotite form at $\sim 1000\ °C$. The sulphide droplets attach to bubbles of devolatilization fluid, which promotes buoyancy-driven transportation in silicate melt. Our study shows that devolatilization fluids can supply large proportions of sulphur and copper in mafic–ultramafic layered intrusion-hosted Cu-Ni sulphide deposits.

[1] Department of Geoscience and Geography, University of Helsinki, Gustaf Hällströminkatu 2, 00014 Helsinki, Finland. [2] Geological Survey of Finland, Vuorimiehentie 5, 02151 Espoo, Finland. [3] Institute of Geochemistry and Petrology, ETH Zürich, Clausiusstrasse 25, 8092 Zürich, Switzerland. [4] Institute of Mineralogy, Leibniz University Hannover, Callinstraße 3, 30167 Hannover, Germany. [5] Department of Geography and Geology, University of Turku, Akatemiankatu 1, FI-20500 Turku, Finland. [6] Earth and Planets Laboratory, Carnegie Institution for Science, 5241 Broad Branch Road, NW, Washington, DC 20015-1305, USA. [7] Department of Chemistry, University of Helsinki, A.I. Virtasen aukio 1, 00014 Helsinki, Finland. [8] Present address: Department of Mineralogy, Institute of Geography and Earth Sciences, Eötvös Loránd University, Pázmány Péter s. 1/C, 1117, Budapest, Hungary.
✉email: ville.z.virtanen@helsinki.fi

Magmas are major sources of heat in the crust in areas of voluminous igneous activity. Accordingly, their wall-rocks are subjected to devolatilization, redox, re-crystallization, and melting. These processes selectively and episodically mobilize chemical components via wall-rock fluids and partial melts, which may subsequently escape the wall-rock and get assimilated by the adjacent magma.

Assimilation of black shales, carbonates, and evaporites can control the carbon and sulphur budgets of mafic magmas[1–3]. External sulphur can saturate the magma in sulphide phases and economically significant base metal deposits form when segregated sulphide melt scavenges, and subsequently accumulates, chalcophile metals from the host silicate melt[2,4]. Wall-rock assimilation influenced the formation of many economically important magmatic Ni–Cu(-PGE) and Cu–Ni(-PGE) deposits such as Noril'sk in Siberia[3], Kambalda in Australia[5], Voisey's Bay in Canada[6,7], Duluth Complex in the USA[1,8–10], and several deposits (e.g. Vammala, Kevitsa, and Sakatti) in Fennoscandia[11,12].

Sulphur mobilization in pyrite-rich ($FeS_2$) wall-rocks can occur when a reducing fluid converts pyrite to pyrrhotite ($Fe_{1-x}S$) with the excess sulphur dissolving in the fluid as $H_2S$[13–16]. This reaction may initiate at temperatures lower than 400 °C in the presence of kerogens, which are common constituents of black shales[13,14]. Thermal breakdown of kerogens produces $CH_4$, which may cause pyrite reduction accompanied by release of $H_2S$[17]. $H_2S$, in turn, has capacity to mobilize economically important base and precious metals, such as zinc, copper, and gold via sulphuric complexes[18,19]. Accordingly, black shales have been suggested as a source of, not only sulphur, but also metals in some magmatic sulphide deposits[11,14,20].

Although many contact aureoles bear evidence of sulphurous fluid formation, notable transport of sulphur and metals to the adjacent magma by fluids has been questioned due to proposed in situ sulphide precipitation within the wall-rock[13,21] and slow rate of diffusive element transport via fluid[7,22]. By contrast, sulphur and metal mobilization is proposed to occur when solid sulphides melt and get transported to the adjacent magma, likely via wall-rock (silicate) partial melt[6,9,10,13,22]. This process is favoured in wall-rock xenoliths and walls of turbulently flowing magma conduits due to the large amount of heat required to reach extensive melting of the wall-rock silicates and sulphides[6,9,10,13,22]. Direct evidence of extensive melting of sufficiently large portions of wall-rock is commonly missing, however, and hidden reservoirs are often envisioned to meet the isotopic mass balance requirements for externally derived sulphur[6,13]. Moreover, detailed studies on the actual mobilization and transport mechanisms within the wall-rocks are lacking, although these form important constraints to the mass balance considerations.

The Duluth Complex hosts several magmatic Cu–Ni(-PGE) deposits with world-class ore resources of ~4.4 Gt at 0.66 wt.% Cu and 0.22 wt.% Ni[1,23,24]. It is an ideal geological setting to study the assimilation process, because most of the sulphur, and possibly undefined amount of base metals, in the deposits were derived from the adjacent Virginia Formation black shale[1,20,25,26]. Sulphur assimilation must have been selective, as evidence from Sr, Pb, and Nd isotopes does not allow bulk assimilation of black shale of more than 5 wt.% for most of the deposits[27,28]. Contrary to bulk assimilation, fluid phase could selectively transport sulphur from the black shale contact aureole to the magma[15,16,28,29] without notably affecting the Sr, Pb, and Nd isotopes. The present understanding is, however, that the fluids have mostly been concealed within the black shale[13,21]. The sulphide deposits are typically associated with abundant black shale xenoliths, some of which are enriched in sulphur and base metals. These xenoliths are the main source for sulphur in the minor norite-hosted deposits[9,10], but cannot explain the more voluminous troctolite-hosted deposits unless some mechanism selectively extracts sulphides from the xenoliths.

Here we constitute a holistic sulphur and copper assimilation model for devolatilizing and partially melting contact aureoles and xenoliths based on black shale heating and partial melting experiments. In these experiments, we used a pristine natural black shale sample from the Virginia Formation (Supplementary Fig. 1) to simulate the devolatilization and partial melting reactions caused by the Duluth Complex magmatism. The conditions estimated from the contact aureole around the Duluth Complex (maximum temperature of 920 °C[30] and pressure of 200 MPa[31]) were used to constrain the experimental conditions (700–1000 °C, 200 MPa). The experiments show that an autochthonous and highly mobile Cu-bearing COHS-fluid (Carbon–Oxygen–Hydrogen–Sulphur-fluid) forms by the devolatilization reactions in silicate subsolidus conditions. The COHS-fluid may not only be directly assimilated, but it can also concentrate sulphur and copper to the proximity of the intrusion. This can lead to local enhancement of sulphur and copper assimilation in places where the S–Cu-enriched wall-rock black shale and xenoliths are in contact with the magma. As the wall-rock black shale melts, the sulphides attach to the devolatilization fluid bubbles, which promotes buoyant migration of the sulphides from the footwall and possibly deeper from the feeder dike system. We present a mass balance calculation showing that the amount of fluids formed within the Virginia Formation contact aureole and xenoliths is sufficient to supply the previously inferred black shale-derived sulphur[1] within the Duluth Complex. This mass balance calculation also provides a tentative quantitative estimate for black shale-derived copper in the Duluth Complex Cu–Ni deposits.

## Results

**Black shale starting material**. The pristine black shale sample (Fig. 1a) used in our experiments was collected more than 10 km away from the Virginia Formation contact aureole (Supplementary Fig. 1) and consequently did not experience contact metamorphism by the Duluth Complex magmas. The Virginia Formation outside the contact aureole has not been subject to temperatures higher than ~400 °C as inferred from the presence of kerogen[13,16]. Our sample is mainly composed of quartz, chlorite, muscovite, and albite, but also contains minor kerogen, titanite, $TiO_2$-phase, pyrite, and chalcopyrite (Figs. 1a and 2a, Table 1). This is the typical phase assemblage outside the contact aureole[31,32], making our sample well-representative of the pre-contact metamorphic rock at the intrusion contact (Table 2). The pyrolysis measurements indicate that the sample contains hydrogen in excess of what can be hosted in $H_2O$ of the hydrous silicates (Supplementary Data 2). The most likely host for the excess hydrogen is kerogen with inferred molar H/C of ~0.3, which is in the range of previously measured kerogen H/C from the same Virginia Formation drill core, where our starting material was collected[33]. The sulphur content of the sample is 0.37 wt.%, which is slightly lower than the average sulphur content of the Virginia Formation black shale (0.60 wt.%[13]). Based on mass balance calculation pyrite grains compose ~95% and chalcopyrite grains ~5% of the sulphides within the sample (Supplementary Eqs. 1.1–1.9).

**Results of the experiments at 700−1000 °C**. All the silicate minerals present in the starting material were also identified in the 700 °C experiment run products (Fig. 1b, Table 1). Additionally, ferrogedrite, a peritectic phase after chlorite dehydration[34,35], was identified. Based on previous experimental studies, chlorite dehydration should be complete at 700 °C[34,35].

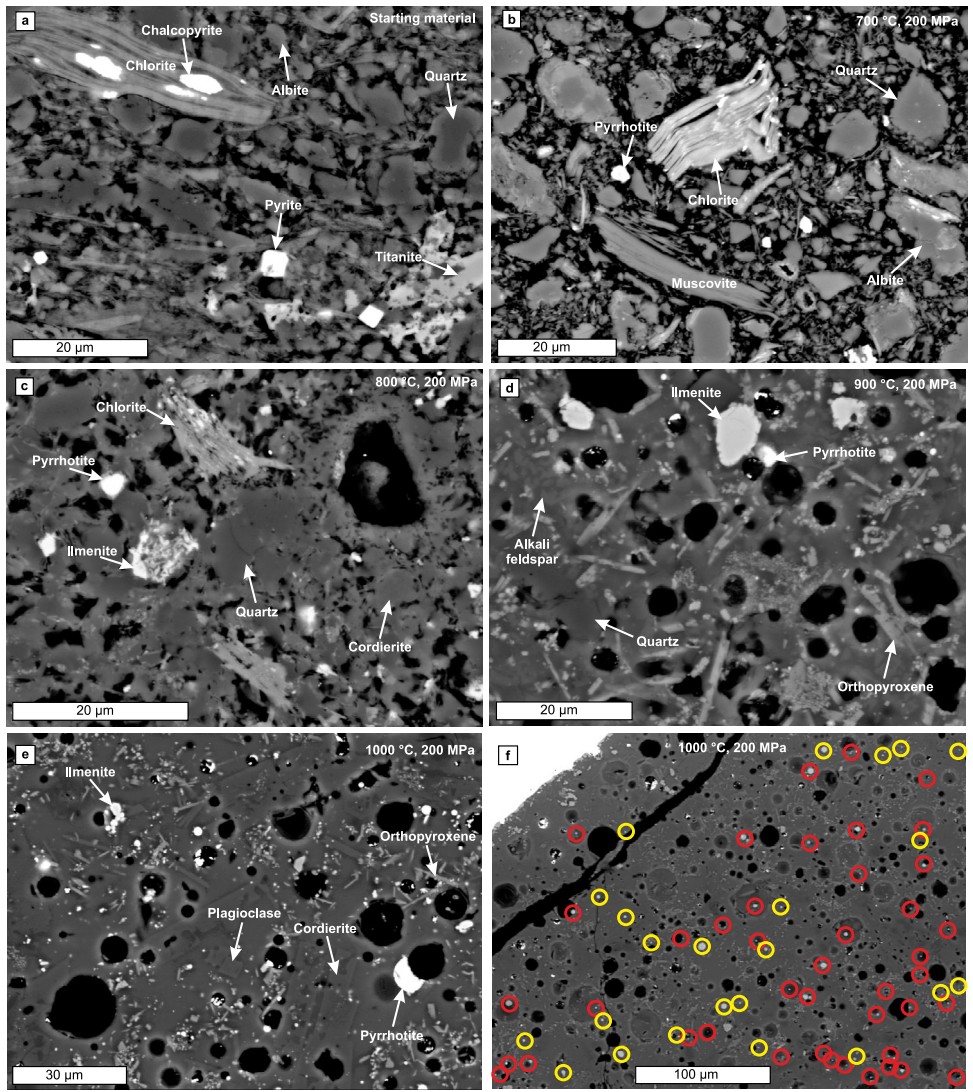

**Fig. 1 Back-scattered electron images showing the textures and phases of the Virginia Formation black shale and the heating and partial melting experiments. a** Starting material (polished thin section) and experiments at **b** 700 °C, **c** 800 °C, **d** 900 °C, and **e** 1000 °C. **f** Distribution of sulphides in the 1000 °C experiment. Red circles ($n = 43$) indicate the sulphides attached to fluid bubbles and yellow circles ($n = 23$) the sulphides that are not observably attached to the bubbles in this 2D section. The black areas in the starting material (**a**) are kerogen and detached grains. The black areas in the **b–f** are vesicles that were filled with fluid during experiment, but which was lost when the capsule was polished open. Pressure in all experiments was 200 MPa.

Hence, the silicate assemblage at the 700 °C experiment is partly metastable due to short experiment duration compared to the reaction rates of the largest grains at near-solidus conditions (see Cho & Fawcett[36]). Kerogen is replaced by disordered graphite (Supplementary Fig. 3) and pyrite and chalcopyrite grains are replaced by Cu-bearing pyrrhotite (Fig. 2b and 3). The incipient replacement of chlorite with ferrogedrite, graphitization of kerogen, and redistribution of copper from minor chalcopyrite to Cu-bearing pyrrhotite indicate the presence of a fluid phase. Consistent pyrrhotite copper content (0.5–1.3 wt.%, Table 1) throughout the capsule volume suggests equilibrium between pyrrhotite and fluid. This underlines that the sulphide-fluid reactions have significantly faster reaction rates compared to the silicate dehydration reactions, which could also lead to similar disequilibrium behaviour in natural systems.

The experiment at 800 °C is characterized by the disappearance of muscovite, albite, K-feldspar, and most of the chlorite (Fig. 1c, Table 1). The coexistence of chlorite, cordierite, ferrogedrite and magnetite (Table 1) indicates that chlorite dehydration in the experiment was incomplete due to slow reaction rates. The mineral assemblage of the run product resembles the mineral assemblages described from the Virginia Formation contact aureole zones with subsolidus devolatilization and incipient partial melting reactions[31] (Table 2). The presence of small amounts of fluid-saturated silicate melt in the run product is inferred indirectly from the presence of rounded vesicles (Figs. 1c and 2c) and the presence of alkali feldspar, cordierite, magnetite, and minor ferrogedrite, which are common peritectic phases of muscovite[37] and chlorite dehydration[34]. The pyrrhotite Cu contents (1.6–8.7 wt.%, Table 1) are higher and less uniform from grain to grain than at the 700 °C experiment (Fig. 2c and 3, Table 1).

Chlorite, ferrogedrite, magnetite, and TiO$_2$-phase disappear between 800 and 900 °C (Fig. 1d, Table 1). The increasing amount of silicate melt and appearance of orthopyroxene correlates with the partially molten zone of the Virginia Formation contact aureole at the proximity of intrusions[31] (Table 2). The range of Cu contents (2.1–3.0 wt.%, Table 1) of pyrrhotite formed at 900 °C (Figs. 2d and 3) overlaps with that of pyrrhotite at 800 °C, but is more tightly constrained (Table 1). In addition to Cu, EDS signal for Ni was detected from half of the measured pyrrhotite

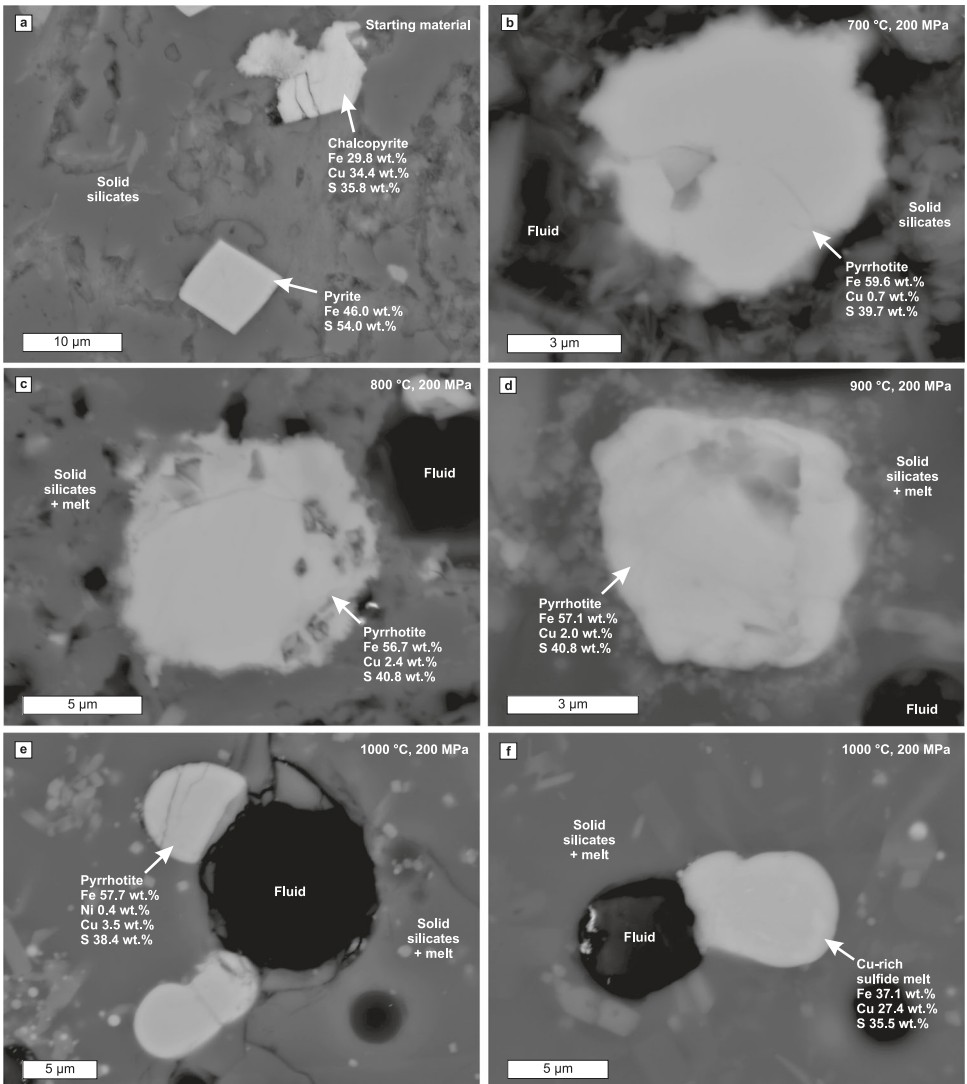

**Fig. 2 Back-scattered electron images of sulphides from the Virginia Formation black shale and the heating and partial melting experiments.** The images are from **a** starting material and experiments at **b** 700 °C, **c** 800 °C, **d** 900 °C, and **e–f** 1000 °C. The compositions (Supplementary Data 3–7) have been measured with energy-dispersive detector of FE-SEM. Note that the "Fluid" indicates empty vesicles, which were filled with fluid during the experiment and that the "Melt" indicates glass that represents quenched melt.

grains in the 900 °C experiment, but Ni contents (0.2–0.6 wt.%, Table 1) were close to or below analytical precision. We report the Ni contents and include them into the normalized pyrrhotite compositions to emphasize that the 900 °C pyrrhotite contains more Ni than pyrrhotite in the lower temperature experiments. In the absence of ilmenite, magnetite, and Ni-bearing sulphides in the starting material, Ni most likely resides in chlorite. At 900 °C experiment, chlorite dehydration melting has completed, allowing Ni to partition to pyrrhotite.

In the 1000 °C experiment, the volume of silicate melt clearly increases relative to the 900 °C experiment, although crystalline silicate assemblage is the same with the exception that alkali feldspar is replaced by more calcic plagioclase (Fig. 1e, Table 1). Additionally, an Fe-oxide phase is present at the proximity of the capsule walls. The cordierite-plagioclase-orthopyroxene assemblage in the 1000 °C experiment run products is observed in the innermost zone of the Virginia Formation contact aureole[31] (Table 2). Three compositionally different sulphide phases are present in the experiments at 1000 °C, here listed in the order of abundance: (1) Cu–Ni-bearing pyrrhotite (Figs. 2e and 3) that has, on average, higher Cu (2.1–4.8 wt.%) and Ni (0.3–1.1 wt.%)

content compared to the pyrrhotite at 900 °C (Table 1); (2) Cu-rich sulphide melt (Figs. 2f and 3) with Cu content of 20.4–27.9 wt.% and Ni content of 0.4–0.9 wt.% (Table 1); (3) Cu-poor sulphide melt that was only identified as a single droplet contains 12.7 wt.% Cu and 1 wt.% Ni (Table 1). Note that the Ni contents are close to or below the analytical precision. All of the sulphides formed at 1000 °C have droplet-like structure and they tend to be concavely attached to fluid bubbles (Figs. 1e–f, 2e–f). Although the structure of the Cu–Ni-bearing pyrrhotite is round, preliminarily suggestive of melt droplets (Fig. 2e), its composition indicates that the phase is actually pyrrhotite solid solution (Fig. 3). Caution should be exercised when interpreting droplet-shaped natural sulphides as representative of a melt phase (see also Botcharnikov et al.[38]). The Fe-oxides and Cu-rich sulphide melt are present at the proximity of capsule walls and their formation is discussed below.

**Evidence for sulphur and copper mobilization via fluid.** In the 700 °C experiment, Cu-bearing pyrrhotite (Fig. 2b) replaces pyrite and chalcopyrite (Fig. 2a) and the Fe–Cu–S ternary shows that

**Table 1 Summary of the phases observed in the Virginia Formation black shale starting material (VF-BS1) and experiment run products at 700, 800, 900, and 1000 °C (VF-BS1.9, VF-BS1.8, VF-BS1.6, and VF-BS1.4).**

| Sample | VF-BS1 | VF-BS1.9 | VF-BS1.8 | VF-BS1.6 | VF-BS1.4 |
|---|---|---|---|---|---|
| Capsule | | $Au_{90}Pd_{10}$ | $Au_{90}Pd_{10}$ | $Au_{90}Pd_{10}$ | $Au_{90}Pd_{10}$ |
| Duration (h) | | 120 | 96 | 72 | 48 |
| T (°C) | | 700 | 800 | 900 | 1000 |
| P (MPa) | | 200 | 200 | 200 | 200 |
| Main phases (>5vol.%)[a] | Quartz, chlorite, muscovite, albite | Quartz, chlorite, muscovite, albite | Quartz, silicate melt, chlorite, cordierite, alkali feldspar | Silicate melt, quartz, alkali feldspar, cordierite, orthopyroxene | Silicate melt, cordierite, plagioclase, orthopyroxene, quartz |
| Minor phases (<5vol.%)[a] | K-feldspar, kerogen, anatase, titanite, pyrite, chalcopyrite, apatite, calcite | K-feldspar, ferrogedrite, rutile, graphite, pyrrhotite, apatite, wollastonite | Ferrogedrite, ilmenite, magnetite, rutile, graphite, pyrrhotite, apatite | Ilmenite, graphite, pyrrhotite, apatite | Ilmenite, Fe-oxide, graphite, pyrrhotite, Cu-rich sulphide melt, Cu-poor sulphide melt |
| Sulphide Cu content (wt.%) | Py: 0.5–1.7 (n = 2), Cpy: 29.7–30.8 (n = 16) | Po: 0.5–1.3 (n = 48) | Po: 1.6–8.7 (n = 38) | Po: 1.5–3.0 (n = 18) | Po: 2.1–4.8 (n = 28), Cu-rich melt: 20.4–27.9 (n = 12), Cu-poor melt: 12.7 (n = 1) |
| Sulphide Ni content (wt.%) | Below detection limit | Below detection limit | Below detection limit | Po: 0.2–0.6 (n = 9) | Po: 0.3–1.1 (n = 28), Cu-rich melt: 0.4–0.9 (n = 3), Cu-poor melt: 1.0 (n = 1) |

For detailed sulphide compositions, see Supplementary Data 3–7.
[a]The abundances were approximated by visual inspection.

the pyrrhotite has less sulphur and copper than the original bulk sulphide (Fig. 3). This indicates that part of sulphur and copper were mobilized via fluid. In order to estimate how much of the sulphur and copper partition into the fluid, we first evaluated whether some sulphur or copper could be hosted by the silicate phases or the experiment capsule. The LA-ICP-MS measurements of the 700 °C experiment capsule walls and interiors indicate that sulphur and copper did not measurably diffuse between the sample material and the capsule (Supplementary Discussion). The role of the silicate phases on the sulphur and copper distribution is discussed below.

Sulphur is incompatible to solid silicates, but can dissolve into silicate melt[39]. We used the rhyolite-MELTS software version 1.2.0[40,41] to model the incipient silicate melt composition at the 700 °C experiment. The resulting composition is $SiO_2$ 70.46 wt.%, $TiO_2$ 0.14 wt.%, $Al_2O_3$ 17.27 wt.%, $Fe_2O_3$ 0.08 wt.%, FeO 0.22 wt.% MgO 0.15 wt.%, MnO 0.14 wt.%, CaO 0.27 wt.%, $Na_2O$ 0.52 wt.%, $K_2O$ 5.98 wt.%, $H_2O$ 4.76 wt.%, and $CO_2$ 0.01 wt.%. Then we used the Smythe et al.[39] model to calculate that the sulphur solubility in the modeled silicate melt is ~50 ppm. Compared to the 3690 ppm of sulphur in the starting material, the sulphur dissolution of 50 ppm is negligibly small, especially given that the amount of partial melt in the experiment was 0–5vol.% based on the phase assemblage (Table 1) and textural observation of the 700 °C experiment products. Sulphur is thus concentrated in the fluid and any sulphur present in the melt is within the uncertainty of the average sulphide compositions used in the mass balance calculations.

Copper is a chalcophile element and it preferably resides in sulphide minerals, but in certain conditions copper can reside in muscovite and chlorite[42], which are present in the 700 °C experiment products. In chlorite, $Cu^{2+}$ can substitute for $Fe^{2+}$ and $Mg^{2+}$ in the octahedral structural site[42]. Limited heterovalent $Fe^{2+}$ and $Mg^{2+}$ substitution is known to take place in muscovite[43], hence it might also incorporate $Cu^{2+}$ in its crystal structure. Instead of $Cu^{2+}$, the reduced $H_2S$-bearing fluid formed in our experiments (see below) is more likely to contain $Cu^+$[19], which is incompatible to chlorite and muscovite. Hence, the silicate minerals can be neglected when considering the mass balance of copper in the 700 °C experiment.

In addition to sulphur and copper distribution, it is important to constrain if iron partitioned into the fluid. To evaluate this, we analyzed a BSE image map of the 700 °C experiment products to determine the pyrrhotite mass (Supplementary Discussion). The results are compatible with the assumption that iron was conserved in the solid sulphides and that no additional iron was introduced from the silicates during the experiment. Based on the above considerations of the sulphur, copper, and iron distribution, we calculated the relative proportions of sulphur and copper in the Cu-bearing pyrrhotite and the fluid in the 700 °C experiment products. The mass balance calculations indicate that the fluid contains ~45 wt.% of the total sulphur and ~60 wt.% of the total copper (Supplementary Eqs. 2.7–2.12).

In order to model the fluid composition, the possible ranges for experimental oxygen fugacity ($f$O$_2$) and sulphur fugacity ($f$S$_2$) have to be estimated. The 700 °C experiment is graphite-saturated (Supplementary Discussion) and thus imposes tight constraints on $f$O$_2$. Using the average Cu-bearing pyrrhotite composition in the 700 °C experiment and the experimentally constrained equations of Toulmin & Barton[44] and Mengason et al.[45], we determined that the experimental $f$S$_2$ was close to −3.8 log units. As the pyrrhotite composition method for $f$S$_2$ is calibrated to 0.1 MPa[44] and because gas fugacities increase with pressure, this represents the lower limit on the experimental $f$S$_2$. The pressure effect for gas fugacities tends to be small; however, e.g. at 700 °C, the log $f$O$_2$ increases ~0.2 log units from 0.1 to 200 MPa based on the equation of Frost[46].

**Table 2 Typical phase assemblages from the pristine parts of the Virginia Formation and from the different zones in the contact aureole[31].**

| Geological context | Pristine Virginia Formation | Hornfels zone | Incipient partial melting zone | Partial melting zone |
|---|---|---|---|---|
| T (°C) | | <670 | 670–740 | 740–870+ |
| P (MPa) | | 200–300 | 200–300 | 200–300 |
| Main phases (>5vol.%) | Quartz, chlorite, muscovite, plagioclase, K-feldspar | Quartz, biotite, cordierite, plagioclase | Quartz, biotite, cordierite, plagioclase, *quartz, K-feldspar*[a] | Orthopyroxene, cordierite, *plagioclase, quartz, K-feldspar, biotite* |
| Minor phases (<5vol.%) | Ilmenite, pyrite, kerogen | Ilmenite, pyrrhotite, graphite | Ti-phase, pyrrhotite, graphite | Ilmenite, pyrrhotite, graphite |

[a]Italicized phases crystallized from melt during cooling.

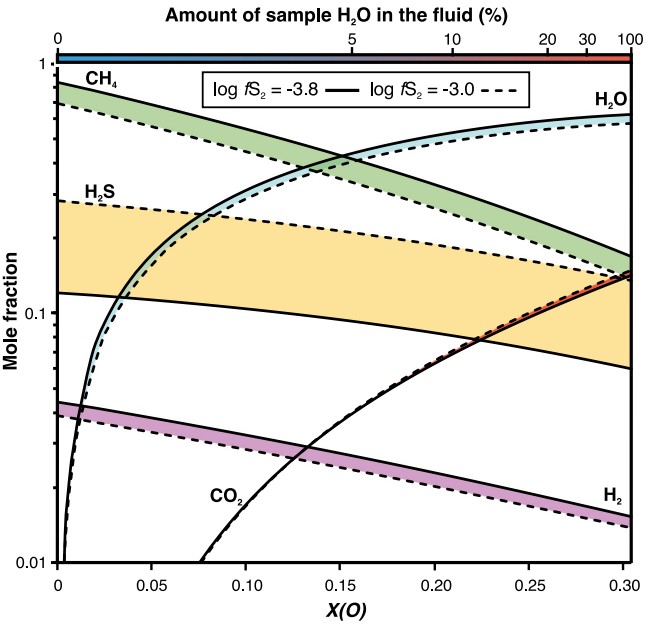

**Fig. 3 Fe–Cu–S ternary diagram showing measured average sulphide compositions of the Virginia Formation black shale starting material and the experimental run products at different temperatures at 200 MPa.** The bulk composition of the starting material is determined with mass balance calculation (Supplementary Eqs. 2.1–2.3). The red stippled projection line starting from the average pyrrhotite composition at 700 °C and going through the bulk sulphide composition indicate the theoretical relative amounts of Fe, Cu, and S in the fluid. The likely range of fluid compositions at 700 °C (red solid line with arrows) has been estimated using Cu/Fe of ≥1, which is typical to $H_2S$ rich fluids at high temperatures[67]. The phase stability fields are from pure Fe–Cu–S experiments of Kullerud et al.[68] at 1000 °C and 1 atm. Only the $Po_{ss}$ stability fields are shown for 900 °C and 700 °C (stippled lines). Note that the stability fields for complex multicomponent natural systems, such as black shales, likely differ to some extent from the pure Fe–Cu–S system. Explanations for the fields: 1. $= Cu_{ss} + Bn_{ss}$, 2. $= Bn_{ss}$, 3. $= Cu_{ss} + Fe_{ss} + Bn_{ss}$, 4. $= Bn_{ss} + Fe_{ss}$, 5. $= Bn_{ss} + M + Fe_{ss}$, 6. $= Fe_{ss} + M$, 7. $= Po_{ss} + M$, 8. $= M$, 9. $= S + M$, and 10. $= S + Po_{ss} + M$. S is sulphur melt, M is Fe–Cu–S melt, $Po_{ss}$ is pyrrhotite solid solution, $Bn_{ss}$ is bornite solid solution, $Fe_{ss}$ is Fe–Cu solid solution, and $Cu_{ss}$ is Cu–Fe solid solution.

To define the fluid composition (Fig. 4), we used the PerpleX thermodynamic modeling software's (version 6.8.9[47]) FLUIDS protocol with the modified Redlich-Kwong hybrid equation of state for COHS-fluids[48]. The fluid speciation was modeled at graphite-saturated conditions and at 700 °C and 200 MPa. To take into account the pressure uncertainty on $fS_2$, we calculated the fluid compositions in the $fS_2$ range between −3.8 and −3.0 log units (Fig. 4). The fluid species mole fractions are modeled against $X(O)$, which is the molar $O/(O + H)$[48]. In our experiment, the $X(O)$ variable is mostly controlled by the initial kerogen-bound hydrogen and $H_2O$ added from dehydrating silicates. The theoretical maximum $X(O)$ of the fluid is ~0.3, when all the $H_2O$ in the sample material resides in the fluid (Fig. 4).

The fluid model shows that the dominant fluid species is either $CH_4$ or $H_2O$ depending on how much $H_2O$ is liberated to the

**Fig. 4 Thermodynamically constrained numerical fluid speciation model for the black shale heating experiment at 700 °C, 200 MPa.** Mole fraction of each fluid species is shown on the y-axis on a logarithmic scale and the x-axis variable $X(O)$ is defined by the molecular $O/(O + H)$ of the fluid. The amount of sample $H_2O$ in the fluid approximates the degree of dehydration and is calculated by adding $H_2O$ to the $H^{kerogen}$. The model was constructed with the FLUIDS protocol of the PerpleX software (version 6.8.9[47]) using the modified Redlich-Kwong hybrid equation of state for COHS-fluids[48]. Oxygen fugacity is buffered by graphite and the sulphur fugacities were approximated using the methods of Toulmin and Barton[44] and Mengason et al.[45].

fluid from the hydrous silicates (Fig. 4). We suggest that sulphur and copper mobilization can occur either via the reaction

$$\text{pyrite + chalcopyrite + kerogen} + H_2O^{\text{dehydration}} \rightarrow \text{pyrrhotite} \\ + \text{graphite} + COHS^{\text{fluid}} + Cu^{\text{fluid}} \tag{1}$$

or possibly without $H_2O$ via

$$\text{pyrite + chalcopyrite + kerogen} \rightarrow \text{pyrrhotite + graphite} + CHS^{\text{fluid}} + Cu^{\text{fluid}} \tag{2}$$

The major sulphur species in the fluid is $H_2S$, which composes 0.06–0.3 mole fractions of the fluid depending on the $fS_2$ and $H_2O$ dilution, whereas the amount of $SO_2$ is negligibly small and below the scale in the Fig. 4. Based on the presence of $H_2S$ in the fluid, we suggest that $Cu^{\text{fluid}}$ forms complexes with $H_2S$, as suggested previously for magmatic-hydrothermal systems (see e.g. Heinrich et al.[18], Zhong et al.[19]).

Many earlier studies suggest that, even if $H_2S^{fluid}$ forms, it reacts in situ with Fe-bearing minerals (e.g. chlorite) to form pyrrhotite, which hinders sulphur transportation via a fluid phase[7,13,21]. In our experiments, the newly formed pyrrhotite is not texturally associated with chlorite, which, in the absence of Fe-oxides, is the only major Fe-bearing phase in the starting material. The mass balance calculations and the Fe–Cu–S ternary (Fig. 3) also suggest that at 700 °C sulphur and copper are not hosted entirely by pyrrhotite, but that they partitioned to the fluid as well.

Pyrrhotite at 800–900 °C has higher copper content than pyrrhotite at 700 °C (Fig. 2c–d, Table 1). As the experiments were conducted in closed capsules, the pyrrhotite Cu-enrichment at 800 °C and 900 °C must come from the Cu-bearing COHS-fluid. This reaction takes place because the ability of pyrrhotite to incorporate copper increases with temperature (Fig. 3). The Cu-enrichment of up to 6 wt.% in pyrrhotite at 800–900 °C suggests that some of the sulphur and possibly all of the copper partition from the fluid to pyrrhotite.

**Formation of Cu-rich sulphide melt and sulphide-fluid composite droplets.** In the 1000 °C experiment, Cu-rich sulphide melt coexists with the Cu–Ni-bearing pyrrhotite. The Cu-rich sulphide melt droplets are present only locally within 300 μm from the capsule wall in areas that are generally depleted in Cu–Ni-bearing pyrrhotite (Fig. 5). This implies that the formation of the Cu-rich sulphide melt is related to reactions occurring only near the capsule walls and causing destabilization of the Cu–Ni-bearing pyrrhotite (Fig. 5a–b). We suggest that diffusive loss of hydrogen through the intact capsule wall, a known phenomenon in high temperature experiments[49,50], caused effective oxidation close to the capsule wall (Fig. 5b). As the process is strongly temperature dependent and because gold is one of the least permeable capsule materials to hydrogen diffusion[49], similar oxidation was not observed in the 700–900 °C experiments.

The oxidation at the 1000 °C experiment caused conversion of Cu–Ni-bearing pyrrhotite to Fe-oxide, which lead to sulphur partitioning to the fluid phase within the narrow zone near the capsule wall (Fig. 5c–d). Copper also partitions to the fluid at 1000 °C as the Fe-oxide grains contain no measurable amounts of copper. When the S–Cu-bearing fluid phase migrates outside the relatively oxidized zone near the wall of the capsule, sulphur and copper partition to the Cu–Ni-bearing pyrrhotite (Fig. 5c). Sulphur and copper partition to the sulphides in ~1:1 ratio, as the measured sulphur to metal ratios of the Cu-rich sulphide melt and Cu–Ni-bearing pyrrhotite are close to equal (Supplementary Data 7). The increased sulphur and copper contents cause the sulphide phase to melt as its composition shifts to the melt-stability field in the Fe–Cu–S ternary (Fig. 3). The relevance of this process to open natural systems is discussed below.

## Discussion
On the basis of the above experimental interpretations and thermodynamic fluid model, we formulate an up-temperature model for magmatic sulphur and copper assimilation from black shales. First, a Cu-bearing COHS-fluid forms within the devolatilizing contact aureole in silicate subsolidus conditions as observed in the 700 °C experiment. The fluid can form by the breakdown of kerogen with or without addition of $H_2O$ from dehydrating silicates (Reactions 1 and 2). This fluid could be assimilated from the footwall adjacent to the intrusion in the early stage of assimilation as well as from xenoliths incorporated into the intrusion (Fig. 6a). Earlier studies suggested that in situ reactions with Fe-bearing phases would immobilize sulphur from the fluid[7,13,21], but our experiments indicate no such reactions.

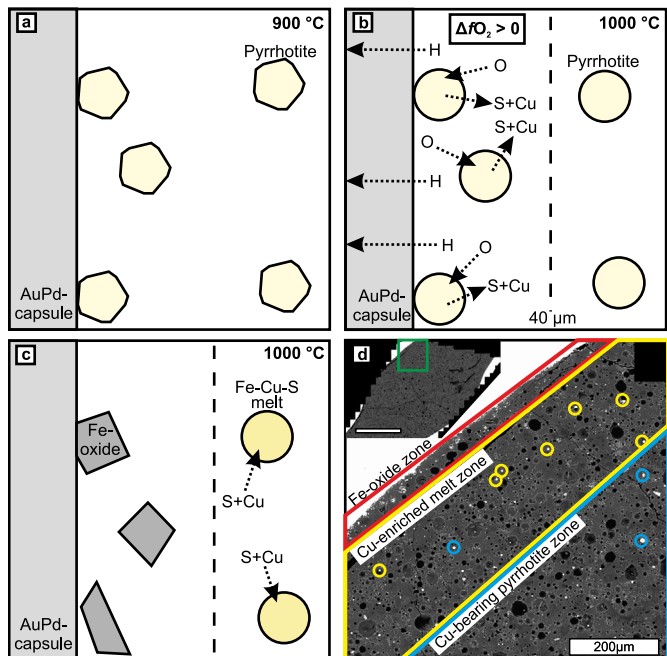

**Fig. 5 A schematic model for the formation of the Cu-rich sulphide melts in the 1000 °C experiment. a** At 900 °C, the Cu-bearing pyrrhotite is stable. **b** At 1000 °C, loss of H through the $Au_{90}Pd_{10}$ capsule wall causes increase in oxygen fugacity in narrow zone close to the capsule wall. The increasing oxygen fugacity increases stability of Fe-oxide and consequently pyrrhotite within the oxidized zone becomes unstable. As the pyrrhotite oxidizes, S and Cu partition into the fluid phase. **c** Fe-oxide replaces pyrrhotite close to the capsule wall. The S–Cu-bearing fluid migrates to the more reduced center of the capsule and reacts with pyrrhotite. The Cu-rich sulphide melt forms when the Cu content of the pyrrhotite increases. **d** Spatial distribution of Fe-oxide grains (red rectangle), Cu-rich sulphide melt droplets (yellow circles), and Cu-bearing pyrrhotite grains (blue circles) within the portion of the 1000 °C experiment capsule (green rectangle) shown in the top left corner (scale bar 1 mm). Only the sulphide grains with measured compositions are indicated with circles.

Opposed to the studies that consider only diffusive transport of sulphur via stagnant fluid network[7,22], numerical modeling and field evidence suggest that crustal fluids physically flow through porosity and fractures[51] and the flow direction is often towards increasing temperature[52,53]. Furthermore, in upper crustal contact aureoles, devolatilization reactions create porosity[51], which increases the possibility of fluid migration from contact aureole towards magma (Fig. 6a). We thus suggest that the fluid phase is able to transport considerable amounts of carbon, sulphur, and copper away from the site of formation—potentially to the intruding magma or hydrosphere and atmosphere.

At 800–900 °C, when a partially molten zone develops in the contact aureole close to the intrusion, the increasing temperature enhances the capability of pyrrhotite to incorporate copper and sulphur (Fig. 6b). This process hinders the assimilation of the S–Cu-rich fluid, while increasingly Cu-rich sulphides form a sulphur- and copper-enriched zone at the proximity of the intrusion (Fig. 6b). Pyrrhotite and Cu-rich sulphide melt can also scavenge small amounts of nickel from the surrounding silicate melt (Fig. 2e, Table 1). If the fluid input for sulphur and copper is sufficiently high and the temperature of the proximal contact aureole reaches ~1000 °C, Cu-rich sulphide melt can form as observed at the 1000 °C experiment (Figs. 2f and 3). Although, the Cu-rich sulphide melt in the 1000 °C experiment forms due to coupled physicochemical reactions caused by the diffusive H-loss

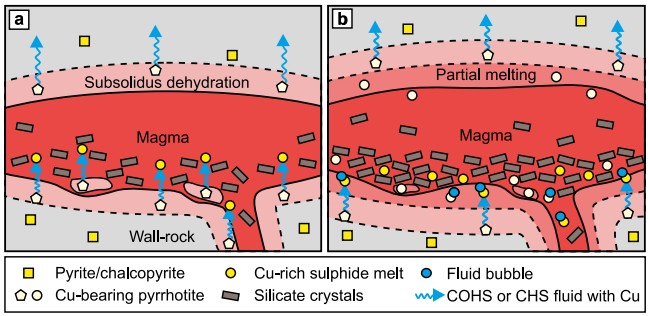

**Fig. 6 Schematic illustration of sulphide mobilization in a developing black shale contact aureole and the modes of assimilation by the magma. a** Devolatilization reactions in the wall-rock footwall and xenoliths form a COHS- or CHS-fluid. The fluid transports sulphur and copper from the footwall and xenoliths to the magma causing sulphide saturation. **b** As the heating of the wall-rock progresses, partial melting zones form close to the intrusion. At this stage, sulphides are suspended in silicate melt and can be assimilated by the magma. In the outer devolatilization zone (similar to that in a) in the footwall, fluids migrate to the partial melting zone forming sulphide-fluid composite droplets. The low-density fluid enhances buoyant ascent of the sulphide droplets suspended in silicate melt in the feeder dike and in the footwall.

from the capsule (Fig. 5), we suggest that it can also form in open natural systems. In footwall black shale rocks, the Cu-bearing COHS-fluid formed by the breakdown of pyrite and chalcopyrite at ≤700 °C could buoyantly migrate from a lower temperature part of the contact aureole towards the higher temperature part closer to the intrusion. As pyrrhotite in the higher temperature part of the contact aureole acquires additional copper and sulphur from the fluid, the Cu-rich sulphide melt forms.

The Virginia Formation black shale contact aureole and xenoliths are indeed locally enriched in carbon, sulphur, and copper[9,10,15,26,54,55]. Former studies explained this enrichment by injection of fractionated Cu-enriched sulphide melt (ca. 20–30 wt.% Cu) from the magma to the black shale[15,26]. Our experiments show that the Cu-rich sulphide melt can form within the black shale by autochthonous fluid refinement and a magmatic source is not necessarily required (Fig. 6b).

The sulphide-fluid composite droplets at the 1000 °C experiment also indicate that some of the pyrrhotite and Cu-rich sulphide melt droplets within the partially molten proximal zone of the contact aureole and xenoliths attach to devolatilization fluid bubbles (Fig. 6b). Similar sulphide-fluid relationships that have been observed in both experiments and in natural systems have revealed that dense sulphides can migrate upwards in less dense silicate melt via bubble flotation[56–59]. Recent computational models have shown that, in favourable conditions, sulphide-fluid composite droplets can also be selectively extracted from partially molten crystal mushes[59]. As partially molten wall-rock is essentially a crystal mush, selective assimilation of sulphur and copper (and minor nickel) via the composite droplets could take place in the melt-dominated regions of the footwall and xenoliths. Such sulphide-fluid droplets are less likely to be assimilated in the roof-side contact aureoles of the intrusion, where the buoyant fluid migration occurs away from the intrusion (Fig. 6b).

In the case of Virginia Formation, extensive partial melting is restricted to ~10 m from the intrusion contact[13], although this could be an underestimate if the intrusion thermally eroded or stoped material from the footwall. In fact, noritic rocks at the footwall contact of the Duluth Complex formed via assimilation of silicate melt from the Virginia Formation black shale[29], which suggests that the zone of extensive partial melting, i.e. the zone where selective extraction of sulphide droplets could have

occurred, was likely wider during the magmatism than the ~10 m observed today. Nevertheless, the spatial extent of possible sulphide flotation is highly localized and considered secondary in importance in sulphide deposit formation compared to the more voluminous assimilation of S–Cu-bearing fluids.

We test the feasibility of our COHS-fluid assimilation model on the basis of mass balance calculations between the contact aureole and the mineralized portion of the Duluth Complex. For these calculations, some generalizations are made: (1) We assume that the fluid does not incorporate iron from sulphides; (2) All the COHS-fluid that forms in the contact aureole volume, migrates to the adjacent intrusion; (3) The contact aureole formed fast enough for the fluid to reach the intrusion during the magmatic stage; (4) The thickness of the contact aureole is constant over the length of the intrusion contact; (5) The measured black shale sulphur and copper contents of 0.37 wt.% and 125 ppm, respectively, are adopted as representative for the average black shale assimilated by the magmas.

Our mass balance calculations indicate that the COHS-fluid hosts ~45 wt.% of the total sulphur and ~60 wt.% of the total copper of the black shale sample used as starting material (Supplementary Eqs. 2.7–2.12). As the sample contains 0.37 wt.% sulphur and 125 ppm copper, the mass of fluid-mobilized elements are 1.7 g and 0.08 g per 1 kg of sample, respectively. The respective masses of the known magmatic sulphide resources in the Duluth Complex are 116 Mt of sulphur and 29 Mt of copper (based on values reported by Listerud and Meineke[23]). These resources are mainly hosted by the Partridge River Intrusion and South Kawishiwi Intrusion (Supplementary Fig. 1).

Adopting the estimate of 87 Mt of total sulphur content of the Duluth Complex assimilated from the black shale (i.e. 75 wt.% of the total sulphur in the sulphide deposits[1]), assimilation of fluids would be required from 52 Gt of wall-rock compositionally similar to our black shale sample. Using 2800 kg/m³ density for the Virginia Formation black shale[60], 52 Gt of rock corresponds to a volume of 18.6 km³, which is the minimum volume for the Virginia Formation contact aureole required for fluid assimilation of sulphur. Based on previously published geological map[30], as well as drill core[33], and structural observations[61] specified below, we evaluated that the size of the Virginia Formation contact aureole exceeds this minimum requirement. For the evaluation, we used the 50 km lateral NE-SW extent of the chain of sulphide deposits[30] (Supplementary Fig. 1) as the along-strike constraint for the volume estimations. The 1.5 km distance between the western margin of the Duluth Complex and the drill hole with observed incipient graphitization of kerogen and pyrite replacement by pyrrhotite within the black shales[33] was used to define the minimum thickness of the heated rock volume (Fig. 7). Projecting the intrusive contact upwards from the surface using the 20° average dip of igneous layering within the overlying Duluth complex[61], we calculated the true minimum thickness of the devolatilized contact aureole to be 500 m (Fig. 7). Using the above values, the required theoretical along-dip length of the rock volume generating the COHS-fluids is 750 m using a simplified orthogonal cross-sectional geometry (Fig. 7). We then adjusted the simplified orthogonal geometry to the pink split quadrilateral area in Fig. 7, which more correctly reflects the downwards-tapering geometry of the Virginia Formation underlying the Duluth complex[62]. The calculated minimum volume is smaller than the inferred size of the Virginia formation contact aureole, which indicates that the formed COHS-fluids could supply the magma with the required amount of sulphur (Fig. 7). The presented estimate requires that the Virginia formation was present at the footwall contact of all of the Cu–Ni mineralizations of the Duluth Complex prior to thermal erosion and xenolith stoping caused by the intruding magma. This has indeed been suggested

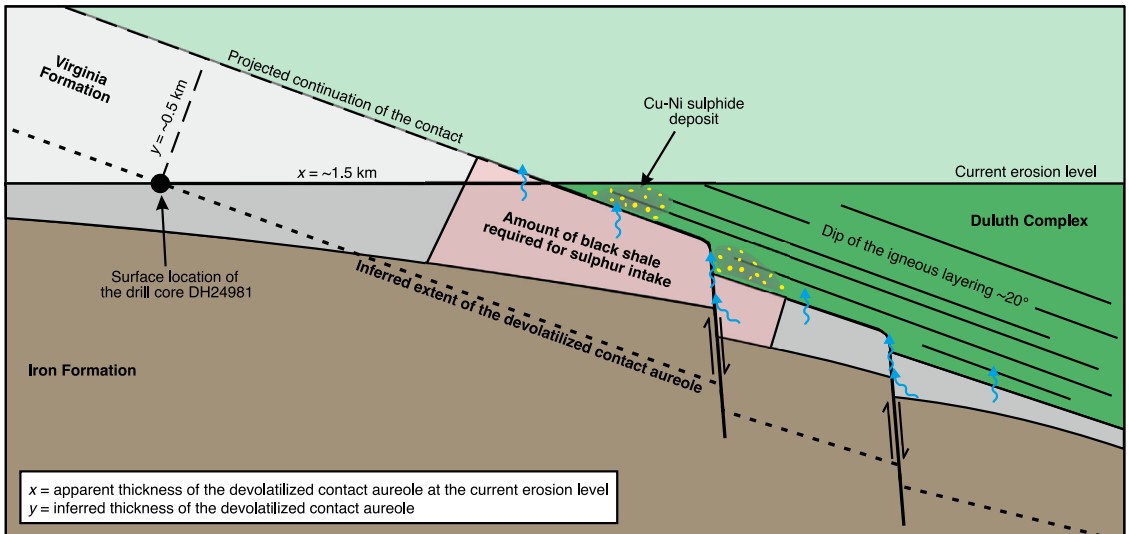

**Fig. 7 In-scale schematic cross-section showing the portion of the Virginia Formation required for the sulphur intake by the Duluth Complex deposits.** The location of the cross-section is indicated in Supplementary Fig. 1. The drill core DH24981 bears evidence of kerogen graphitization and replacement of pyrite by pyrrhotite[33], indicating devolatilization and sulphur mobilization. The intrusive contact was projected upwards from the surface using the 20° dip of the Duluth Complex igneous layering[61], providing true thickness of 0.5 km for the approximated devolatilized contact aureole. The thickness of the contact aureole exceeds the present thickness of the downwards-tapering Virginia Formation below the Duluth Complex[62]. The proportion of the Virginia Formation required for the sulphur intake by the Duluth Complex Cu–Ni sulphide deposits is equal to a 500 × 750 m rectangle and smaller than the size of the Virginia Formation within the contact aureole. Note that the illustration does not take into account possible sulphur mobilization from xenoliths and the walls of the feeding channel of the intrusion.

by Miller et al.[24], and is further supported by the presence of the black shales, comparable to the Virginia Formation along the Duluth Complex contact in the NE (Supplementary Fig. 1).

Based on the above, the suggested fluid transport mechanism is in excellent agreement with the indications from sulphur isotopic data[1,8]. The suggested fluid dominance is also concordant with the isotopic data showing generally limited assimilation of Sr, Pd, and Nd[27,28] that are concentrated in silicate minerals and melts. The mass of copper assimilated from the black shale is 4 Mt, which corresponds to 15 wt.% of the total copper in the Cu–Ni deposits of the Duluth Complex. This substantiates that black shales can act as considerable sources for copper in magmatic sulphide deposits. The amount of mobilized copper, however, depends on relative amounts of sulphides, kerogen, and $H_2O$ and on the consequent $fO_2$ and $fS_2$ in the wall-rock and the $H_2S$ activity in the fluid phase. Hence transposition of our results to other systems requires modeling of these reactions and fluids for the given geological environment. Copper isotopic studies could provide more insight into this subject.

We emphasize that the suggested devolatilization reactions and the formation of COHS-fluids are not necessarily restricted to the level of present contact zones of the sulphide deposits if fertile wall-rock material was available already along the pathways of the intruding magmas. Our experiments and mass balance calculations nevertheless show that selective assimilation of fluids enriched in sulphur and copper is likely the dominant mode of assimilation over that of sulphide melt, at least in the case of black shale wall-rocks.

## Methods
**Experimental procedures**. The natural Virginia Formation black shale starting material comes from a ~50 cm piece of the split drill core MDDP2 (548449 E, 5259726 N, UTM, zone 15) from the depth of ~480 m. It was recovered from the Department of Natural Resources Drill Core Library, Hibbing, Minnesota. Detailed description of the entire drill core can be found in Pfleider et al.[63], where the MDDP2 is referred as Hole 2. A geological map showing the drill core location can be found in the Supplementary Fig. 1.

The sample was crushed with a jaw crusher at the Department of Geosciences and Geography, University of Helsinki, and pulverized with an agate ball mill (mesh size 90 μm) at the Geo Labs Ontario (Geo Labs method code SAM-AGM). To reach finer grain size necessary to reach faster reaction rates for the experiments, further pulverization was performed with a Retsch MM200 horizontal agate ball mill (45 min) and agate mortar (70 min) at the Institute of Geochemistry and Petrology, ETH Zürich. The final grain size was mostly ≤10 μm but some ≤40 μm grains of mainly quartz and flaky chlorite persisted through the pulverizing. The sample powder was dried at 120 °C overnight before it was immediately sealed in $Au_{90}Pd_{10}$ capsules (2.3 × ~6 mm), which were weighed before and after each experiment to ensure that the capsule stayed intact. The sample masses within the capsule volume were in the range of 15–20 mg.

The estimated metamorphic pressure for the Virginia Formation contact aureole is 200 MPa[31] and the highest metamorphic temperature found in the adjacent granitoid wall-rocks is 920 °C[30] (Supplementary Fig. 1). Accordingly, we performed experiments at 200 MPa pressure and at temperatures of 700, 800, 900, and 1000 °C. Oxygen fugacity ($fO_2$) in the experiments was inherently buffered by graphite and all of the experiments were inherently fluid-saturated (Supplementary Fig. 5). The experiments were conducted in $Au_{90}Pd_{10}$ capsules using externally heated molybdenum-hafnium-carbon (MHC) pressure vessels with Ar gas as pressure medium at the Institute of Geochemistry and Petrology, ETH Zürich. In this experimental setup, part of the pressure vessel resides inside an electric furnace, which generates a 20 mm hotspot zone with the desired experiment temperature inside the pressure vessel. The other end of the vessel is exposed to room temperature. The pressure vessel and the furnace are attached to an axle, which allows 180° tilting of the entire setup.

The experiments were started by sliding the capsule to the hotspot zone of the vessel inside the furnace. Nova Swiss diaphragm compressor was used to pressurize the vessel with Ar. Based on a predetermined temperature-dependent Ar expansion inside the vessel, the target pressure was reached by first pressurizing the vessel to an intermediate state and then by heating to the target temperature, which simultaneously increased the pressure inside the vessel to the desired 200 MPa. Temperature was monitored with a K-type Ni–Cr thermocouple manufactured by Omega and pressure with a WIKA pressure gauge. The accuracy of the measured experiment temperatures is 1–2% and the pressure deviated no more than 1 MPa from the target pressure of 200 MPa during the experiment. Isobaric quenching was performed by tilting the pressure vessel from subhorizontal 10% inclination to vertical orientation, which caused the capsule to fall to the cold end of the pressure vessel allowing an estimated cooling rate faster than 100 °C/s. Experiment duration was 48 h at 1000 °C and was increased by 24 h per −100 °C to compensate for slower reaction rates at lower temperatures. Failed experiments are shortly reviewed in the Supplementary Notes.

**Whole-rock measurements of the starting material**. The whole-rock major element oxide, trace element, and volatile compositions of the starting material

were measured at the Geo Labs Ontario and the data is available in the Supplementary Data 1. The major element oxide composition was measured from a fused borate glass bead with X-Ray Fluorescence (XRF) method (Geo Labs method code XRF-M01). The accuracy of the XRF standard measurements is good as deviation from the standard material is less than 3% for major element oxides other than MnO, for which the deviation is 10%. Coefficients of variation ($C_v$) are generally <2% with the exception of MnO for which the $C_v$ is 6% (Supplementary Data 1).

For whole-rock trace element analyses the sample was first dissolved by Closed Vessel Multi-Acid Digestion (Geo Labs method code SOL-CAIO) and then measured with inductively coupled plasma mass spectrometry (ICP-MS; Geo Labs method code IMC-100). Quality control measurements deviate from the certified standard composition by ≤5% in the case of elements other than Hf, Th, Pr, and Er for which the deviation from the standard is <10% and Mo, Cs, Tl, Tb, and Tm for which the deviation is <23% (Supplementary Data 1). For most of the elements, $C_v$ is ≤5%, but for Ni, Mo, Be, and Tl it is ≤9%.

For the whole-rock FeO content measurements of the starting material, the sample was dissolved in a non-oxidizing acid mixture. The amount of FeO in the solution was then determined using potentiometric titration with permanganate standard solution (Geo Labs method code FEO-ION). Deviation from standard is 0.3% and $C_v$ 1% (Supplementary Data 1).

The total free and crystalline $H_2O$ and the total sulphur contents of the whole-rock were measured by infrared absorption of combustion products in oxygen-rich atmosphere (Geo Labs method codes: IRW-$H_2O$ for $H_2O$ and IRC-100 for S). Accuracy of the $H_2O$ measurements cannot be evaluated due to lack of certified standard material. The $C_v$ for $H_2O^+$ and $H_2O^-$ are 2.4% and 15.4%, respectively. For sulphur, deviation from the standard material is 0.1% and $C_v$ is 2.1% (Supplementary Data 1).

The total organic and inorganic carbon measurements were performed at the Core Laboratories at Houston with Rock-Eval 7 analysis. Stepwise pyrolysis (25 °C/min) at 300–600 °C is first used to thermally decompose the pyrolyzable organic carbon. The decomposed hydrocarbons are measured with flame ionization detector and $CO_2$ and CO are measured simultaneously with infrared cells. Inorganic carbon and residual organic carbon are combusted in oven in oxidizing atmosphere at 300–850 °C (30 s isothermally at 300 °C, followed by continuous heating by 20 °C/min). Deviations from the standard values for total organic and inorganic carbon are 0.6% and 1.9%, respectively. $C_v$ is 4% for both organic and inorganic carbon (Supplementary Data 1).

The C, H, N, and S contents of the sample VF-BS1 were measured with a HANAU vario Micro cube automatic elemental analyzer (Elementar Analysensysteme GmbH, Germany) at the Department of Chemistry, University of Helsinki. For the measurement, 5 mg of sample material was first heated to 105 °C overnight to remove possible moisture. Then the sample material was closed gastightly and the excess air was removed. The sample was then combusted in a single step at 1150 °C in oxygen-rich atmosphere. The $NO_x$ and $SO_3$ were reduced to $N_2$ and $SO_2$ in a reduction tube at 850 °C. The reduced combustion products, $H_2O$, $CO_2$, $SO_2$, and $N_2$ were collected in adsorption column before flushing them to a thermal conductivity detector (TCD) with He gas. $N_2$ was measured immediately, which was followed by the consecutive release of other combustion products to the TCD by heating: $CO_2$ at 60 °C, $H_2O$ at 140 °C, and $SO_2$ at 210 °C. Based on three standard material measurements, deviation from the standard value is ≤0.16% and the $C_v$ is ≤1.9% for all the elements. As C and S were also determined with other methods, only the H content of the sample material is reported in the Supplementary Data 1. The measured total C content (1.24 wt.%) is within ~2% compared with the combined organic and inorganic C content (1.21 wt.%) measured by the Core Laboratories. The measured S content is 3770 ppm, which is within ~2% from the S content of 3690 ppm measured by the GeoLabs.

**In situ sulphide measurements**. For analysis of the experiment products, the recovered capsules were mounted in epoxy, ground open, and polished perpendicular to their longest dimension after which they were coated with carbon. Sulphide chemistry of the starting material and experiment run products were analyzed at the Geological Survey of Finland, Espoo using a JEOL JSM-7100F Field Emission Scanning Electron Microscope (FE-SEM) equipped with an Oxford Instruments energy-dispersive (EDS) X-Max 80 mm$^2$ detector. FE-SEM was used as the grains are too fine-grained (<5 μm in general; Fig. 2a–f) to be reliably analyzed with a conventional electron microprobe with thermionic emission source, e.g. tungsten filament. The FE-SEM analyses were performed with an accelerating voltage and probe current of 20 kV and 1.3 nA, respectively and with 10 s measuring time. For the sulphide measurements, pure element standards were used for Fe, Ni, and Cu, and $FeS_2$ for S.

Certified marcasite and chalcopyrite standards were also measured with the FE-SEM during analysis sessions and served as internal standards. We compared our measured marcasite and chalcopyrite compositions with the reported standard compositions measured in-house at the Geological Survey of Finland, Espoo with Cameca Camebax SX100 electron microprobe analyzer (EMPA) equipped with five wavelength dispersive detectors. $C_v$ for the measured marcasite and chalcopyrite standards are ≤0.6% (Supplementary Data 8). The sulphide measurements from the starting material and most of the experiments show good precision with $C_v$ ≤ 1.2% for Fe, Cu, Ni, and S (Supplementary Data 3–7). For Fe, Cu, and S in the

Cu-bearing pyrrhotite at the 800 °C experiment and the Cu-rich sulphide melt at the 1000 °C experiment, $C_v$ are ≤2.9% (Supplementary Data 5). The larger $C_v$ reflects rather heterogeneity among the individual sulphides than true analytical precision.

Based on the marcasite and chalcopyrite standard measurements, we calculated correction factors for Fe, Cu, and S concentrations according to the differences between the measured and reported standard compositions. These correction factors (all <2%) were applied to the measured sulphide compositions of the starting material and the experiments (Supplementary Data 3–7). As the standards did not contain Ni, we applied the correction factor of Cu to Ni for the Cu–Ni-bearing pyrrhotite at the 900 °C experiment.

Raman spectroscopy was used to identify ferrogedrite (700 °C experiment) and graphite (all experiments). We used an NT-MDT Ntegra confocal Raman spectrometer equipped with a 532 nm frequency doubled Nd:YAG laser and an Andor Newton CCD detector (cooled to −60 °C) at the Department of Chemistry, University of Helsinki. We used a ×100 magnification objective for all the measurements, which provides spatial resolution of ~1 μm and grating was set to 1800. Silicon standard was used for spectrometer calibration before the sample measurements. The graphite spectra in the 700, 800, 900, and 1000 °C experiment run products (Supplementary Fig. 3) were measured for five seconds with two accumulations and ferrogedrite in the 700 °C experiment run products (Supplementary Fig. 4) was measured for 30 s with two accumulations. In order to avoid possible contamination from carbon coating, we measured the Raman spectra from a non-coated surface of the sample material that was removed from the experiment capsules (Supplementary Fig. 2).

**Experiment capsule composition measurements**. The analyses of the $Au_{90}Pd_{10}$ capsules were performed with a Coherent GeoLas MV 193 nm laser ablation system attached to an Agilent 7900 s quadrupole ICP-MS at the Department of Geosciences and Geography, University of Helsinki. The measured data are listed in the Supplementary Data 9. The capsules were ablated with a laser diameter of 60 μm and the standards with a laser diameter of 90 μm at the measured surface. Ablation frequency was set to 10 Hz and laser fluence to 10 J/cm$^2$. The depths of the ablation pits were measured to be ~20 μm with a Leica DM2700 P optical microscope. Helium was used as carrier gas with the flow rate of 1050 ml/min. Argon plasma gas flow in the ICP-MS was 850 ml/min and the RF power was 1500 W. We used a single detector with 10 ms dwell time for each measured mass ($^{29}$Si, $^{34}$S, $^{56}$Fe, $^{60}$Ni, $^{63}$Cu, $^{65}$Cu, $^{105}$Pd, $^{107}$Ag, and $^{197}$Au).

Background was measured for 40 s and sample for 50 s, which was followed by a washout of 30 s. The NIST-610 silicate glass was used as the external standard and NIST-612 silicate glass as the internal standard[64]. The standards were measured before the samples, after 20 sample measurements, and after all the samples were measured. Si was used as the external standard for the NIST-612[64]. The relative differences between the measured average masses in the NIST-612 and the reference composition[64] are: $^{34}$S 21.5%, $^{56}$Fe 4.26%, $^{60}$Ni 1.0%, $^{63}$Cu 8.9%, $^{65}$Cu 0.66%, $^{105}$Pd 34.3%, $^{107}$Ag 6.1%, and $^{197}$Au 4.71%. The $C_v$ are: $^{34}$S 3.8%, $^{56}$Fe 33.8%, $^{60}$Ni 8.3%, $^{63}$Cu 0.6%, $^{65}$Cu 1.3%, $^{105}$Pd 1.9%, $^{107}$Ag 1.8%, and $^{197}$Au 2.4%. For internal standardization of the experiment capsules, we used the average Au content measured from 15 $Au_{90}Pd_{10}$ capsules (90.9 wt.%, $1\sigma = 0.14$, $n = 255$) with a JEOL-JXA 8200 EMPA at the Institute of Geochemistry and Petrology, ETH Zürich[65]. The SILLS software version 1.2.1d[66] was used for data reduction.

## Data availability
The authors declare that the data supporting the findings of this study are available within the paper and its supplementary files.

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

## Acknowledgements

Dean Peterson and the staff of the Hibbing Drill Core Library of the Minnesota Department of Natural Resources, Lands, and Minerals Division are acknowledged for helping in acquiring the black shale sample. Giuliano Krättli, Radoslaw Michallik, and Heikki Suhonen are thanked for helping with sample preparations, Severi Juttula, Markus Metsälä, Christoph Beier, and Adam Abersteiner for helping with the analytical procedures, and Fabio Cafagna, James Connolly, and Sanni Turunen for discussions. The LA-ICP-MS data are contribution #1 from the Environmental and Mineralogical Laboratories (Hellabs) of the Department of Geosciences and Geography, University of Helsinki. The study was funded by the Academy of Finland Grants 295129 (J.S.H.), 306962 (V.J.V. and J.S.H.), and 327358 (V.J.V.). Open access funded by Helsinki University Library.

## Author contributions

V.J.V. conducted the experiments, part of the thermodynamic modeling, most of the analytical work (FE-SEM, LA-ICP-MS, Raman spectroscopy), and wrote most of the manuscript. J.S.H. and F.Molnár developed the original research idea and collected the sample. M.W.S. designed the experimental procedure. F.Marxer contributed in designing and conducting the partial melting experiments. P.S. developed the structural interpretation and drafted Fig. 7. N.K. conducted the thermodynamic modeling for graphite stability. K.M. contributed in designing and conducting the pyrolysis analyses. All authors contributed in interpreting the results and writing the manuscript.

## Competing interests

The authors declare no competing interests.
