## [Peer Review File · Nature Communications]

Fluids as primary carriers of sulphur and copper in magmatic assimilationReviewers' comments:

Reviewer #1 (Remarks to the Author):

The authors present an experimental study to quantify the assimilation of sulfide from black shales by silicate melt. Background is that in many sulfide bearing intrusions S isotopes suggest that much of the magmatic sulfide had gone through a sedimentary cycle before it was assimilated by the melt. Much of this is intuitive. When hot mafic melts ($> 1150^{\circ}\text{C}$) come in contact with C-H-O-S volatile rich material, the volatile inventory will be mobilised. Some part will be assimilated by the melt to form magmatic sulfide mineralisations, some other part may escape by degassing.

The authors reacted black shales at 700 to 1000°C and 2 kbar for 48 plus x hrs. in Au-Pd capsules. The idea is to quantify by partial melting of shales if and to what extent shale sulfides can be assimilated by mafic intrusions like Duluth. The experiments were only partly successful:

(1) At 700 and 800°C, was equilibrium reached? I doubt this. Why should chlorite survive 800°C (Fig. S2c)? Chlorite is stable in metapelites to $\sim 650^{\circ}\text{C}$ (check the AFM diagrams). I suspect that run times were too short. With respect to run times the authors should consult the experiments performed in the past to calibrate the garnet-biotite exchange thermometer - weeks.

(2) The phases in Fig. S2c still are fragments. Were the gas pressures (2 kbar) at 800°C sufficient to cause the Au-Pd capsules to collapse?

(3) Little experimental details are presented. How was temperature measured? Should the authors decide to publish their study elsewhere they are advised to elaborate on the experimental details since their technique is not so well established as piston cylinder experiments.

(4) The texture of Fig. S2a - what does this image show? Is it a natural texture or a powder composed of fragments? Polished section images of natural shales look quite different. What is the black matrix material around the fragments - voids? One should not publish such obscure BSE images, then claim (or imply) the experiments were successful and equilibrium was reached.

(5) It seems there was a massive redox gradient at run conditions - not surprising when pressure is imposed by Ar which is never oxygen free. Near the capsule contact sulfide is consumed by oxidation - not really encouraging for experiments that wish to model a natural process involving metal transport by H₂S (cf. eqn. 1).

(6) Was Cu absorbed by the Au-Pd capsule? Did the authors analyse the capsule material for Cu prior to and after equilibration?

(7) Were the experiments fluid saturated? The way to test this is to pierce a capsule after the run under binoculars and check if water escapes.

(8) The rounded mss they observe was crystalline at run conditions. Many equilibrium sulfides and metals form rounded, not euhedral crystals. The authors should check Chadwick, G. A. (1972). *Metallography of Phase Transformations*. London: Butterworths, 302 pp. Besides, if a copper rich sulfide is coexisting, one phase (here mss) must be crystalline because no miscibility gap is known in Fe-Ni-Cu-S.

(9) It is speculated rather prominently on the role of kerogens to mobilise wall rock S which I find interesting. However, no evidence is presented by the experiments that this is indeed the case - all what the authors say is based on assumptions presented by previous work that I did not check. Given how oxidized the experiments are (sulfide oxidation) I would be surprised if TOC did survive and H₂S formed. In what form were the kerogens present in the starting shale? Graphite (which is not a kerogen)? Do the authors find any evidence to support that eqn. 1 played a role in their experiments? No fluids were analysed.

(10) When the authors pierced their experimental capsules, did they encounter any H₂S smell (cf. eqn. 1)? I doubt they did because the experiments were rather oxidised. Or did they think about

testing this?

I thought papers published by Nature Comm. should communicate a major progress in solving problems that were previously considered unsolved. I don't think this is the case here. The editors of the journal should have realised this by themselves. The assimilation of S and other volatiles, when hot basaltic melts come in contact with volatile rich shales, is not a fundamental problem, at least I never perceived it as such. S isotopes of magmatic sulfides show that sedimentary S assimilation must be commonplace. Given that much of the relevant information is found in the supplementary - which is longer than the main text - I suggest that this ms be submitted to a standard petrologic journal with main text and supplementary integrated.

I have not checked the validity of the mass balance calculations 1 and 2 (supplementary information). I did not see much sense in this when the authors try to solve a problem with their experiments that does not really exist. Besides, the scales (capsule size vs intrusion size) they are trying to connect by their modelling are too large for any meaningful quantification.

Some detailed comments keyed to line numbers below

75 sulfide ores close to footwall contact, not near the roof

113 did you see kerogens in any form in the starting shale? Or only inferred from 1.08 wt.% TOC?

115 I understand the fluid after experiment was not analysed for Cu? Fluid saturation after quenching, did the authors check if fluid escaped when capsules were opened? Did they smell H₂S (cf. eqn. 1)?

140 rounded shapes of pyrrhotite do not necessarily indicate they were melt, see Chadwick, G. A. (1972). Metallography of Phase Transformations. London: Butterworths, 302 pp. The fact that Cu rich melt was coexisting indicates that mss was solid even when it is rounded.

142 how quickly were the charges quenched? As fast as in piston cylinder experiments?

147 I understood the wall rocks exposed are granitoids? cf. 75, better state clearly if and where black shales are exposed and what they look like near the contact.

151 why, on what basis? Why not similar in-situ reactions as in your experiments?

180 has this influence of kerogens on S mobilisation been shown experimentally, or is this statement an assumption based on previous work?

188-195 I am not well acquainted with that technique. How much material (mg) do the Au-Pd capsules accommodate?

364 why do the authors need to calculate the composition of the starting material? See the excel table submitted as supplementary.

385 why do you need ascent of sulfide by flotation? Previously you stated the sulfide mineralisation occurs at the footwall contact with granitoids.

Fig. 1 rather poor quality for a field emission probe.

Fig. 2 is standard. This type of Cu distribution is typical for all magmatic sulfides, natural and experimental. I adds nothing to the problem.

Fig. 3. This figure is confusing. Why not draw 4 isothermal sections at 700 to 1000°C using FeS - CuS - S as the corners of an equilateral triangle? The region below the FeS-CuS join is not of interest for you. Where are the sulfide solidi located at those temperatures?

Fig. 4 is a simple cartoon that could be drawn without the experiments presented here.

Reviewer #2 (Remarks to the Author):

The authors performed partially melting experiments on black shale from Paleoproterozoic Virginia Formation at 200 MPa and temperature range from 700 to 1000°C. The experimental results demonstrate that fluid phase mobilizes significant amount of carbon, sulphur and copper upon dehydration reaction at the contact aureole between the wall-rock and rising mafic magma. The authors speculate on the role of Cu-rich sulphide melts in mobilization of the copper and sulphur.

I have two main issues with the paper.

First, I am not convinced by the chosen experimental approach.

Conducting experiments to understand the behaviour of Cu in sulphur-bearing systems at magmatic conditions are not easy. The commonly used capsule materials, including one used in

this study react very easily with both Cu and S. It is well known fact (e.g. Zajacz et al 2001, Tattitch and Blundy 2015). Therefore, I am very sceptical about the mass balance calculations and as a result the conclusions of this study.

How do you know that your Cu is not allowing with the capsule bellow 1100 °C (at 1100° authors acknowledge this problem in the supplementary material) and some of the Cu lost from the original composition not to the present fluid but rather to the capsule wall? If it is the case it will affect the mass balance calculations.

What was the fO_2 of the experiments? Any control/estimates?

L91-134 this part of the paper is based around the dehydration reaction 1 and 2 which happen at 700°C. However, at 800 and 900 °C you have much more complex reactions which involve muscovite (important water bearing phase). I would anticipate change in water activity, fH and fO in the system. All these parameters are crucial for sulphur speciation and metal transport. None of this is discussed in the paper, apart from the caution given at the end of the manuscript L180-185.

The second issue with the paper is that without detail reading of the Supplementary information the context of the main text is lost.

L86-89 Summarise well my comment above.

Although the experimental approach in understanding of ore deposits is very important the format of the paper would be more suitable for another journal which does not have restriction for the length of the main text, number of references and visuals.

Minor comments

Fig1 -SEM images are very poor quality (with 3µm scale). Reduce magnification if it is not possible to achieve better focus at the current one.

On image c, pyrrhotite is in the centre and what is the surrounding phase with irregular shape?

Fig2 – Is Cu in wt% or ppm?

L118-119 is it “potentially all carbon” or as “large proportion of carbon, sulphur and copper” repetition?

Compositions normalised to 100 can be misleading, especially for melts, it would be beneficial for the reader to see also data before the normalisation to estimate true Cu content of the sulphide liquid and as well as melt composition itself.

Reviewer #3 (Remarks to the Author):

Review of Nature Communications 243616_0

General comments. I have always been resistant to the idea that metamorphic fluids could be assimilated by magmas, moving up temperature gradients and possibly also fighting pressure gradients surrounding degassing magmas that are expelling fluid with significant overpressure above ambient lithostatic pressure. Instead I've favored bulk assimilation of the whole rock instead of selective transfer of S into magmas. This article is the first to show me a convincing mechanism for the addition of S- and metal-bearing fluid to a cooling magmatic intrusion. For this reason, I think it is an important piece of work and it should be published.

However there are some problems that should be corrected before the work can be accepted. These involve:

- burial of the most interesting ideas and the over-arching story in the supplementary files. Move these front and centre and build the discussion so that it provides the scaffolding required to show

why these ideas have merit

- failure to address fluid composition
- failure to address all possible contributions to the mass balance

The whole section from L137 suffers from the lack of a simple description of the mechanism proposed for transfer of metals in a S- and Cu-rich fluid from the lower parts of the thermal aureole up to a hotter zone where these metals can be added to sulfide melt and then perhaps further elevated, with the attached sulfide droplets, to the base of the intrusion where they may be assimilated. I get the picture (I think) having read the supplement first, but the ordinary reader will never see the supplement where the story is really fleshed out. This really needs to be changed so that readers can easily grasp what is being proposed. The most interesting ideas are totally buried in the supplement where only the nerdiest of the nerds will ever find them.

There is a lot of literature out there on C-O-H-S fluids and it should be consulted to strengthen the part of the paper concerned with fluid composition. Alternatively, the authors could throw up their hands and say that the fluid composition is unknowable because there are insufficient constraints, and the article would still have documented a transfer of Cu from solids to this undefined fluid, and the rest of the story might still hold water. pardon the pun. I would prefer to see either a solid discussion of fluid composition or an admission that all that can be guessed at is the approximate Cu:S:C ratio of this fluid.

The mass balance in the supplementary file only considers Cu and S distribution between sulfide phases and fluid, totally neglecting the possibility that both of these elements may enter silicate melt or that Cu may enter silicate solids (they do note in the text that Ni partitions into silicates). The mass balance looks nice but it is possibly incomplete, and it needs to be buttressed by some very forceful arguments why silicate melt and solids can be ignored, or modified to take them into account explicitly.

In general I think we should avoid referring to minerals in plural, like "pyrrhotites" instead of calling them "pyrrhotite grains". We don't refer to "quartzes". Perhaps the editor won't agree or care. It's not important.

Main text:

I read the main text after the supplementary file, so the comments might seem a little out of sequence. Please bear with me. I don't have time to go back and rewrite the entire review, so please just keep this sequence in mind as you read the remarks I make.

L13 sulphide phases is not a compound noun and should not be hyphenated

L53 assimilation of wall rock does not necessarily require that it first melt, then mix and hybridize with the mafic magma. Wall rock can also be dissolved in the mafic magma, which is a different process and much easier to accomplish. Think of sugar in tea. The sugar is not melting. It is dissolving, at a temperature far below that at which sugar melts or caramelizes.

L76 mineralization should not be pluralized

L94 please spell out orthopyroxene on first use of the abbreviation opx: ... orthopyroxene (opx).. : anyway, why abbreviate one mineral name but no others? I think there is enough space to write the whole mineral name. opx is jargon.

L104. Ok here is the equation I was missing in the supplementary file. Understood. Note my comments there that there might also have been significant amounts of unbound H₂O in the abundant porosity visible in the SEM image of the starting material. Please comment on this possibility, and if necessary add it to your mass balance

L109 is there no CO in these carbonic fluids? I don't recall whether CO is stable at the CH₄-CO₂ boundary at moderate pressures. Note however that in low pressure industrial applications in settings like smelters the oxidation of reduced carbon tends to generate large amounts of metastable CO, which is a potent reductant, even though the equilibrium species might be CO₂. I have no idea whether this could be important. I'm just putting it out as something to consider and perhaps to comment on in the text.

L116. I'm not satisfied with this rather vague discussion of fluid composition. What is this fluid, exactly? Is it a carbonic fluid dominated by CH₄-CO₂ and perhaps also some CO? Is there any H₂O in it? If it is a carbonic fluid, what possible relevance does the work of Heinrich on aqueous fluids

have? Do sulfide complexes as envisioned by Heinrich in saline aqueous fluids exist in carbonic fluids? Perhaps the metals are transported as carbonyl complexes in the carbonic fluid (this is how Ni is refined – we know it works). Carbonic fluids are non-polar solvents utterly unlike aqueous fluids, which have high dielectric constant and promote solvation of cations and anions. They become more non-polar at high temperatures and their miscibility with carbonic fluids increases but they are still different.

The Cu-Fe-S ternary doesn't have C or O or H in it, so how can it be used to estimate the composition of a C-O-H-S fluid? The fluid-pyrrhotite tie-lines in Fig 3 are, as far as I can see, fabricated to illustrate a point based on an argument rooted in a mass balance that I have pointed out below is very poorly constrained, rather than being constrained by theory or measurement. I think the authors need to say so very clearly so readers won't mistake this bit of poetic licence for fact. It might well be true but the evidence is circumstantial.

See also my concerns about mass balance – how much H₂O might there be in this fluid?

L141. I have to admit that at first I looked at the figure and thought the pyrrhotite was a quenched liquid. But with a little careful consideration I have to agree with the authors. However I think they need to stress this point a little more carefully. They might, if there is space, cite a pair of papers by Botcharnikov et al. that described melting experiments with pyrrhotite and sulfide melt in them where they were forced to admit in a second paper that some of their putative sulfide melt globules in the first paper were actually solid pyrrhotite that just liked to have a round smooth crystal habit. If the authors don't stress this, then some readers will think they have missed something important.

L156. If the authors care and have space for extra citations they might also cite LeVaillant et al., (2017: PNAS) for further discussion of the compound drop phenomenon in mafic magmas. There are also two upcoming papers in JPet and in EPSL by Yao and Mungall digging much deeper into the physics of compound drop transport in crystal-free magmas and crystal mushes but these may not be published before the present manuscript. These articles are probably not necessary to cite but the authors might find them interesting or useful.

Supplementary file:

L54 Why is the capsule in the hot end elevated by 10° even during the run? Doesn't this run the risk of it being displaced slightly by any vibration, and then not being reliably held in the hot end? Is this a typo?

L62 The interpretation is reasonable. Was there no detectable Cu in the capsule? I suppose that if these runs are being discarded it doesn't matter too much. Note that if Cu was truly alloyed with AgPd then it would be present in solid solution. If it is forming an intermetallic compound with fixed stoichiometry like skærvaardite then it is not an alloy, strictly speaking, but it should be easier to detect with an SEM.

L66. Geolabs also does a wet chemical titration for ferrous iron so that ferric/ferrous ratio can be determined (I don't think it is the metavanadate method, but it is similar). But perhaps this is not critically important. I'm reading the supplement first, before I see what was done with the results.

L113. marcasite probe standard – this must be tricky to work with. Doesn't it oxidize and blow up the polished mount? Why not pyrite? not relevant to the paper. I'm just intrigued.

L136. I don't understand. What reactions are being invoked to support the assertion that volatilization of kerogen requires inputs of H₂O? Can't oxygen be provided by reduction of ferric iron (for which you don't have any data) to produce carbonic fluid of CO-CO₂? If you don't know what was the composition of the fluid then how can you be sure that it had H₂O in it? Also, was there no H in the hydrocarbons to begin with? This bit of text is too incomplete to leave as is. Then we see that pyrite-pyrrhotite reaction has liberated half of the S in the starting material, so what is the speciation of S in the fluid? It should be possible to estimate, at least approximately, the bulk composition of this fluid and estimate speciation using one of many available fluid speciation models from the internet. There may be constraints on fO₂ from the silicate and oxide mineral assemblage and there should be a constraint on fS₂ (at least an upper bound if pyrite is completely absent since you are still below the terminal breakdown temperature for pyrite

melting). Again, maybe after I read the main text I will decide that this is irrelevant. See also comments on SEM images – how much unbound pore fluid was there prior to heating, and what was its role in the melting and transport processes?

L177. The discussion of Ni transport from silicates into sulfide melt or solid is apt. I'm not sure about Cu, and this also leads me to wonder if the Ni story is complete. What about the possibility of Cu transport from chalcopyrite into silicate crystals or melt? Have any of the silicate minerals or the glass been analyzed for Cu and Ni? The partition coefficients for Cu and Ni from silicate melt to sulfide solids may be rather small. If sulfide melt has not yet appeared, but silicate minerals are beginning to melt, then the Cu content of the silicate melt may not be negligible.

Mass balances: See comments above regarding possible distribution of Cu and Ni into phases other than fluid or sulfides. Also – why can Fe not have been incorporated into oxide or silicate minerals? Fe-rich metamorphic silicates commonly form in abundance surrounding sulfides that have lost S during metamorphic devolatilization.

Formation of the Cu-rich sulfide melt:

I like the idea of oxidation by H₂ loss through the capsule walls. The subsequent ideas about degassing of a S-Cu-rich fluid from the cooler end of the metamorphic thermal gradient and its ascent into the hotter zone is intriguing and worth examining. I'm not sure why this important concept is relegated to the supplementary information. I hope I will see it again in the main text.

Estimation of the feasibility...

The mass of Virginia Fm over 27.5 km of strike and an unknown dip extent seems like the wrong thing to be calculating. If we assume that the lower contact of the intrusion against the Virginia Fm was essentially planar, and much larger than the snippet that is preserved today at the current erosional level, it seems to me that the valuable thing to calculate is the amount of Virginia Fm present in an arbitrary vertical column of rock, let's say 1 m square and spanning the vertical thickness of the thermal aureole. This can be compared with the amount sulfide present in a typical 1 m squared vertical column of the intrusion itself to see if the amount derived from the base of the complex is similar to the amount of S in the mineralized zone.

Supplementary Figures

S1 looks fine.

S2: a. there is a lot of porosity in the starting material. How much trapped pore fluid was present prior to experiment (or prior to metamorphism) and what role would this presumably H₂O-rich fluid have played in the melting process and the putative transport of fluid and sulfide melt?

b. There is only one Cpy crystal in panel a. How do you know that there was not Cpy or Py in the 700 °C run product? I expect that your search was probably careful enough but it would help to support the assertion that Cpy and Py were gone if you could explain how much surface area was searched before concluding that they were completely absent.

Supplementary data:

I would suggest that if the analytical data for sulfides is reportable at 3 significant digits, then the normalized molar quantities and ratios should also be reported at three significant digits. Reporting mole fractions of Cu of either 0.0 or 0.1 is not very informative. Furthermore, the data in the right-most columns is labeled as (A.%), but they are definitely not percentages. They seem to have been normalized to something but it is unclear how the normalization was done since the sum of metals and sulfur is quite variable, but the amount of each individual element also varies.

Reviewers' comments:

Reviewer #1 (Remarks to the Author):

The authors present an experimental study to quantify the assimilation of sulfide from black shales by silicate melt. Background is that in many sulfide bearing intrusions S isotopes suggest that much of the magmatic sulfide had gone through a sedimentary cycle before it was assimilated by the melt. Much of this is intuitive. When hot mafic melts (> 1150°C) come in contact with C-H-O-S volatile rich material, the volatile inventory will be mobilised. Some part will be assimilated by the melt to form magmatic sulfide mineralisations, some other part may escape by degassing.

The authors reacted black shales at 700 to 1000°C and 2 kbar for 48 plus x hrs. in Au-Pd capsules. The idea is to quantify by partial melting of shales if and to what extent shale sulfides can be assimilated by mafic intrusions like Duluth. The experiments were only partly successful:

(1) At 700 and 800°C, was equilibrium reached? I doubt this. Why should chlorite survive 800°C (Fig. S2c)? Chlorite is stable in metapelites to ~ 650°C (check the AFM diagrams). I suspect that run times were too short. With respect to run times the authors should consult the experiments performed in the past to calibrate the garnet-biotite exchange thermometer - weeks.

It is true that silicate equilibrium was not reached at 700 and 800 °C, and we failed to address this matter clearly regarding the 800 °C experiment in the original submission. In the revised manuscript, we have addressed the partial silicate metastability in lines 120–124 in the case of the 700 °C experiments and in lines 132–134 in the case of the 800 °C experiments. In the case of the 700 °C experiment, we also state in lines 128–130 that, whereas the silicates are in metastable state, the sulphide, graphite, and fluid are in mutual equilibrium due to faster reaction rates. It is true that the silicate equilibrium would change the fluid equilibrium, which would then possibly change the sulphide equilibrium as well. However, we would also like to point out, that in nature it is very much possible that the fluids and sulphides equilibrate and the fluid leaves the system before the silicates reach equilibrium.

We also performed Raman spectroscopy on the 700 °C experiment products and identified ferrogedrite spectrum, which implies that the submicron chlorite experienced dehydration during the experiment. The presence of ferrogedrite at the 700 °C experiment is now mentioned in the manuscript lines 119–120 and the Raman measurements are presented in the Supplementary Information section “Identification of graphite and ferrogedrite using Raman spectroscopy”.

(2) The phases in Fig. S2c still are fragments. Were the gas pressures (2 kbar) at 800°C sufficient to cause the Au-Pd capsules to collapse?

Yes, the capsules collapsed in all of the experiments. We are not quite sure what is meant by the “fragments”, but if it refers the black angular areas, they were fluid-bearing cavities that were not able to acquire rounded shape because of such low degree of melting.

(3) Little experimental details are presented. How was temperature measured? Should the authors decide to publish their study elsewhere they are advised to elaborate on the experimental details since their technique is not so well established as piston cylinder experiments.

Externally heated pressure vessel techniques have been well established since their invention in late 1940's by O. F. Tuttle. We feel that the specific experimental technique used in this study has been explained in detail in the original manuscript. However, the reviewer is right that we failed to include the type of thermocouple that was used for temperature measurements. In the revised version, we state that the thermocouple is a K-type Ni-Cr thermocouple manufactured by Omega and that the pressure was measured with a WIKA pressure gauge (lines 441–443). We also added a reference (Tuttle, 1949) for anyone interested in the details of the experiment technique (lines 440–441).

We would also like to point out that piston cylinders are not applicable to experiments at pressures as low as 200 MPa.

(4) The texture of Fig. S2a - what does this image show? Is it a natural texture or a powder composed of fragments? Polished section images of natural shales look quite different. What is the black matrix material around the fragments - voids? One should not publish such obscure BSE images, then claim (or imply) the experiments were successful and equilibrium was reached.

The Fig S2a is a BSE image from a polished thin section, which we failed to state in the figure caption. The texture then is the natural texture. The black areas around the silicates were formed partly by grains detached during polishing and partly by kerogen. In the revised version, we have stated in the caption of the figure 1a (line 800–801) that the BSE image is taken from a polished thin section.

(5) It seems there was a massive redox gradient at run conditions - not surprising when pressure is imposed by Ar which is never oxygen free. Near the capsule contact sulfide is consumed by oxidation - not really encouraging for experiments that wish to model a natural process involving metal transport by H₂S (cf. eqn. 1).

The oxidation effect is only present at 1000 °C experiment (and possibly in the failed experiment at 1100 °C mentioned in the Supplementary Information). In other experiments there is no preferred oxide occurrence or absence of sulphides close to the capsule walls. The oxidation is actually not a result from oxygen addition from the Ar gas, because oxygen is not able to diffuse through noble metal capsules. Before experiments the capsules were welded shut and we know that the capsules stayed intact during the experiments as the capsules were weighed before and after every experiment, as we stated in the original manuscript (now line 432). Hydrogen, however, is able to diffuse through solid noble metals. The diffusion rate is highly temperature dependent and we did not observe any oxidation in the 700–900 °C experiments.

We apologize for not being clear about this in the submitted manuscript. There's now a detailed description of the process in the revised version (lines 273–291). In regard to H₂S, in the original manuscript we only discussed H₂S at 700 °C, which was not subjected to hydrogen loss and consequent oxidation.

(6) Was Cu absorbed by the Au-Pd capsule? Did the authors analyse the capsule material for Cu prior to and after equilibration?

In the original manuscript, we had not measured the Au-Pd capsules for contamination from the sample. To make sure that the capsules did not react with the sample material we performed a set of additional LA-ICP-MS measurements from the capsules. These measurements show that there was no measurable contamination in any of the experiments. In the revised version, these measurements are shortly mentioned in lines 174–177, and the reader is directed to the Supplementary Information, where the issue is discussed in detail in the sections "Analysis of Fe-S-Cu diffusion in the experiment capsule using LA-ICP-MS". The LA-ICP-MS method is described in the manuscript methods section lines 546–568 and the necessary sample preparation in Supplementary Information section "Identification of graphite and ferrogdrite using Raman spectroscopy".

(7) Were the experiments fluid saturated? The way to test this is to pierce a capsule after the run under binoculars and check if water escapes.

All of the experiments were fluid saturated, which we did not explicitly state in the originally submitted manuscript, although the information was available as we mentioned the presence of a fluid phase in the case of all the experiments. In the revised version, we now state explicitly in the lines 436-438, that all the experiments were fluid-saturated.

(8) The rounded mss they observe was crystalline at run conditions. Many equilibrium sulfides and metals form rounded, not euhedral crystals. The authors should check Chadwick, G. A. (1972). Metallography of Phase Transformations. London: Butterworths, 302 pp. Besides, if a copper rich sulfide is coexisting, one phase (here mss) must be crystalline because no miscibility gap is known in Fe-Ni-Cu-S.

The reviewer is correct, and we already clearly stated it in the main text of the original submission ("At 1000 °C, this phase is round, suggestive of melt droplets (Fig. 2e), but its composition indicates that the phase is pyrrhotite solid solution", now lines 266–268). We have also expanded the subject in the revised version, where we refer to a pair of papers, where droplet-like sulphide was first suggested to be melt, but in the later paper it was corrected that the phase was actually solid (see lines 268–269).

(9) It is speculated rather prominently on the role of kerogens to mobilise wall rock S which I find interesting. However, no evidence is presented by the experiments that this is indeed the case - all what the authors say is based on assumptions presented by previous work that I did not check. Given how oxidized the experiments are (sulfide oxidation) I would be surprised if TOC did survive and H₂S formed. In what form were the kerogens present in the starting shale? Graphite (which is not a kerogen)? Do the authors find any evidence to support that eqn. 1 played a role in their experiments? No fluids were analysed.

As stated above, the 700 °C experiment was not subjected to oxidation (due to hydrogen loss). The ubiquitous presence of kerogen in the not contact metamorphosed Virginia Formation has been verified by previous researchers (e.g., Ripley et al., 2001). In the original manuscript, we stated that the kerogen has low H/C based on previous studies, but as we performed new pyrolysis measurements, we found out that the sample contains hydrogen in excess of what can be hosted by H₂O, which suggests that the H/C of the kerogen is actually ~0.3. This is now mentioned at lines 107–112. The pyrolysis method is described in lines 488–504.

(10) When the authors pierced their experimental capsules, did they encounter any H₂S smell (cf. eqn. 1)? I doubt they did because the experiments were rather oxidised. Or did they think about testing this?

We did not perform capsule piercing after experiments. Instead, as stated in the methods, we grinded the capsules open after mounting them to epoxy (now lines 506–508). The grinding was performed so that a sand paper was constantly under running water.

We performed new Raman spectroscopy measurements to determine that graphite (after experimental kerogen graphitization) was stable phase in all the experiments, which is stated in the revised manuscript lines 436–438. These results are available in the revised version of Supplementary Information section “Identification of graphite and ferrogdrite using Raman spectroscopy” and the Raman spectroscopy method is described in the manuscript (lines 534–545). Special sample preparation was conducted to be sure that carbon coating was not accidentally measured (see “Identification of graphite and ferrogdrite using Raman spectroscopy” in Supplementary Information).

We used this information to constrain the oxygen fugacity during the experiments. We also constrained the sulphur fugacity in the experiments based on the metal to sulphur ratios of the Cu-bearing pyrrhotite at the 700 °C experiment (lines 210–213). With these new information, we were able to produce a thermodynamically constrained fluid speciation model with PerpleX software, which shows that H₂S is certainly the main sulphur-bearing fluid species (lines 218–241).

I thought papers published by Nature Comm. should communicate a major progress in solving problems that were previously considered unsolved. I don't think this is the case here. The editors of the journal should have realised this by themselves. The assimilation of S and other volatiles, when hot basaltic melts come in contact with volatile rich shales, is not a fundamental problem, at least I never perceived it as such. S isotopes of magmatic sulfides show that sedimentary S assimilation must be commonplace. Given that much of the relevant information is found in the supplementary - which is longer than the main text - I suggest that this ms be submitted to a standard petrologic journal with main text and supplementary integrated.

We also think that the assimilation of sulphur and volatiles from wall-rocks by basaltic magma is not a novel idea and it was not our point to prove that. Our point, and the new discovery here, is the mechanism that forms the fluid that mobilizes sulphur and that the fluid also transports copper during early stages of wallrock heating. This has not been proven previously as it is extremely difficult to conclude with certainty from natural rocks that were subject to open system behavior (chemical exchange). For this reason there is an ongoing debate over whether the dominant mode of assimilation was by fluid and/or melt. Without knowledge on how these elements are mobilized and assimilated, it's virtually impossible to produce meaningful mass balance models on how much could have been assimilated from the wall-rock or released to atmosphere.

The manuscript was initially submitted to Nature Geoscience, which has much smaller word limit than Nature Communications. We did not modify the manuscript to the Nature Communications format as we used the manuscript transfer portal as suggested by Rebecca Neely, the Editor of Nature Geoscience (“Nature Communications is the Nature Research flagship Open Access journal. If you would like this work to be considered for publication there, you can easily transfer the manuscript by following the instructions below. It is not necessary to reformat your paper”).

In the revised manuscript, we have taken advantage of the much larger word limits of the Nature Communication and moved much of the information initially in the Supplementary Information to the body of the manuscript.

I have not checked the validity of the mass balance calculations 1 and 2 (supplementary information). I did not see much sense in this when the authors try to solve a problem with their experiments that does not really exist. Besides, the scales (capsule size vs intrusion size) they are trying to connect by their modelling are too large for any meaningful quantification.

The mass balance calculations in the supplementary information are related to the distribution of elements between phases within the capsule. The calculations show how much of S and Cu of total amount in the capsule occur in the solid sulphides and fluid.

The problem we try to solve is what is the mechanism that mobilizes S and Cu, when black shales are heated. As this mechanism is not well known, it is very difficult to try to quantify the magmatic assimilation of these elements. Since the magmatic assimilation of these elements is a key process in the formation of Ni-Cu sulphide mineralizations, we think that there was a critical issue that we solved with our experiments. With the experiments, we can also quantify the amount of S and Cu mobilized at the different stages during heating. We have not encountered a single study that would look at the mobilization at multiple temperatures and consider the different processes. We feel that these are extremely important points for anyone, who tries to understand the magmatic assimilation of S and Cu or release of C, S, and Cu to atmosphere in more shallow systems.

We do not agree that the small scale of the capsule cannot be extended to larger scales. The amount of material (mm scale capsule vs. km scale contact aureole) does not change the thermodynamics and mass balances of the reactions that inevitably take place as temperature increases – this is the well sound basis for all

experimental petrology. Groundbreaking experiments that have revealed processes that function in the scales of planetary bodies have been performed in tiny capsules.

Some detailed comments keyed to line numbers below

75 sulfide ores close to footwall contact, not near the roof

We are not sure what this comment refers to, but we guess that it means that in the original manuscript, the sentence was too complicated to understand. In the revised version, we removed this information as it is not extremely necessary and might cause unnecessary confusion among readers.

113 did you see kerogens in any form in the starting shale? Or only inferred from 1.08 wt.% TOC?

The presence of kerogen instead of graphite can be inferred from the pyrolysis results with hydrogen in excess of that what could be derived purely from H₂O. In the reflected light optical microscopy, the kerogen is present as interstitial irregular light gray masses, which are present throughout the thin section. We can also identify kerogen from the RockEval analysis spectrum as kerogen breaks down earlier in the pyrolysis than graphite and also releases small amount of hydrogen. Optical microscopy and the RockEval spectrum are not shown in the paper, though, as we feel that there's enough evidence of the presence of kerogen already. Additionally, kerogen has been identified from the same drill core earlier (lines 109–112)

115 I understand the fluid after experiment was not analysed for Cu? Fluid saturation after quenching, did the authors check if fluid escaped when capsules were opened? Did they smell H₂S (cf. eqn. 1)?

We did not check the fluid saturation by piercing the capsule, but inferred it from the porous texture and redistribution of Cu. We did not try to smell the capsule when opened. We are also quite skeptical of using smelling test as a qualitative method of research.

There is now a thermodynamic model of the fluid composition in the revised version (lines 218–243, Fig. 4). This model shows that H₂S is stable and most abundant sulphurous species in the fluid.

140 rounded shapes of pyrrhotite do not necessarily indicate they were melt, see Chadwick, G. A. (1972). *Metallography of Phase Transformations*. London: Butterworths, 302 pp. The fact that Cu rich melt was coexisting indicates that mss was solid even when it is rounded.

As we already clearly stated in the original submission, pyrrhotites are solid, but the Cu-enriched phase is melt. We have tried to be even more clear about this in the revised version (lines 265–272 of the revised manuscript).

142 how quickly were the charges quenched? As fast as in piston cylinder experiments?

The quench rate is "faster than 100 °C/s" as stated in the supplementary information, where the method was described in detail in the originally submitted manuscript. Quench rate in piston cylinders is distinctly slower than in the externally heated pressure vessels. In the revised version, the quenching procedure is now explained in the manuscript method section lines 446–449.

147 I understood the wall rocks exposed are granitoids? cf. 75, better state clearly if and where black shales are exposed and what they look like near the contact.

We have now simplified this and do not mention the granitoids anymore as they caused confusion (lines 61–74).

151 why, on what basis? Why not similar in-situ reactions as in your experiments?

We have discussed this process in detail in the revised manuscript lines 273–291.

180 has this influence of kerogens on S mobilisation been shown experimentally, or is this statement an assumption based on previous work?

This is cautionary statement for workers who might use our results to quantify the S and Cu mobilization in another contact aureole. It is highly likely that the amount of Cu in the fluid depends on the amount of H₂S in the fluid, which depends on the relative amounts of kerogen, H₂O, and sulphides in the wall-rock (lines 405–407). It does not influence the main findings of the study.

188-195 I am not well acquainted with that technique. How much material (mg) do the Au-Pd capsules accommodate?

The capsules can accommodate 15–20 mg of sample material. This is now stated in line 432.

364 why do the authors need to calculate the composition of the starting material? See the excel table submitted as supplementary.

We calculate the bulk sulphide composition in order to track and visualize the S and Cu distribution among phases in the experiments in the figure 3 and also to calculate the mass balance for S and Cu mobilization via fluid.

385 why do you need ascent of sulfide by flotation? Previously you stated the sulfide mineralisation occurs at the footwall contact with granitoids.

It is not necessarily “needed”, but this is something that we observed and hence could occur in nature as well. It is often stated that assimilation would take place deeper in the feeder dyke system, but then the problem is to transport the dense sulphides from there in suspension in less dense silicate melt. The fluid-sulphide composite droplets would have much lower effective density than sulphide droplets by themselves. Similarly, it would be more difficult for sulphide droplets to end up in the magma from the footwall without the aid of the fluid.

We have discussed this process in more detail in lines 337–354 of the revised manuscript and added references that have suggested this process can take place in magmatic systems.

Fig. 1 rather poor quality for a field emission probe.

Unfortunately, the first author was not able to produce better quality BSE images and due to the corona virus situation the FE-SEM is currently not available for use. For this reason, we cannot get new BSE images. Still the quality is better than would be expected from a traditional thermionic filament-sourced based SEM at these magnifications.

Fig. 2 is standard. This type of Cu distribution is typical for all magmatic sulfides, natural and experimental. I adds nothing to the problem.

This figure has been removed from the revised manuscript.

Fig. 3. This figure is confusing. Why not draw 4 isothermal sections at 700 to 1000°C using FeS - CuS - S as the corners of an equilateral triangle? The region below the FeS-CuS join is not of interest for you. Where are the sulfide solidi located at those temperatures?

We agree that, this figure might look confusing at first to some readers, especially as so many things can be seen from it. We are hesitant to draw 4 isothermal sections of the diagram, however, as only the narrow area of the diagram close to the Fe-S binary is needed for 700–900 °C experiments, and separate sections would require unnecessary space. In our opinion, the sulphide solidi for 700 °C and 900 °C are quite clearly shown with stippled lines and the 1000°C solidus with solid line in the enlargement of the Fe-Cu-S ternary on the right. Solidus at 800 °C is not showing as it was not determined in the Kullerud et al. (1969), which we used as a reference.

Fig. 4 is a simple cartoon that could be drawn without the experiments presented here.

We feel that the simple cartoon helps to visualize how the process could occur in nature. The details in the figure are based on observations from the experiments, i.e., that the fluid forms in the subsolidus dehydration zone (Fig. 6a), and that at later stage this fluid reacts with the sulphides in the partially molten zone and forms Cu-rich melt and composite sulphide-fluid droplets (Fig. 6b).

Reviewer #2 (Remarks to the Author):

The authors performed partially melting experiments on black shale from Paleoproterozoic Virginia Formation at 200 MPa and temperature range from 700 to 1000°C. The experimental results demonstrate that fluid phase mobilizes significant amount of carbon, sulphur and copper upon dehydration reaction at the contact aureole between the wall-rock and rising mafic magma. The authors speculate on the role of Cu-rich sulphide melts in mobilization of the copper and sulphur.

I have two main issues with the paper.

First, I am not convinced by the chosen experimental approach.

Conducting experiments to understand the behaviour of Cu in sulphur-bearing systems at magmatic conditions are not easy. The commonly used capsule materials, including one used in this study react very easily with both Cu and S. It is well known fact (e.g. Zajacz et al 2001, Tattitch and Blundy 2015). Therefore, I am very sceptical about the mass balance calculations and as a result the conclusions of this study.

How do you know that your Cu is not allowing with the capsule bellow 1100 °C (at 1100° authors acknowledge this problem in the supplementary material) and some of the Cu lost from the original composition not to the

present fluid but rather to the capsule wall? If it is the case it will affect the mass balance calculations.

LA-ICP-MS was used to measure the capsule compositions and based on these measurements, there was no significant loss of Cu or S to the capsule from the sample material in any of the experiments (see Supplementary Information sections "Analysis of Fe-S-Cu diffusion in the experiment capsule using LA-ICP-MS". In the revised manuscript, we mention that we considered the the capsules in lines 174–176. We also considered the possible effects of silicate phases to the Cu and S mass balance with the conclusion that silicate phases play no significant role in the distribution of these elements at 700 °C experiment.

What was the fO_2 of the experiments? Any control/estimates?

We performed Raman spectroscopy measurements and identified graphite in all of the experiment run products (see Supplementary Information section "Identification of graphite and ferrogedrite using Raman spectroscopy". This is contrary to what we stated in the original manuscript, where we stated that all carbon was in the fluid phase. In the revised version of the manuscript, we have stated that the experiment fO_2 is buffered by graphite (lines 436–438).

L91-134 this part of the paper is based around the dehydration reaction 1 and 2 which happen at 700°C. However, at 800 and 900 °C you have much more complex reactions which involve muscovite (important water bearing phase). I would anticipate change in water activity, fH and fO in the system. All these parameters are crucial for sulphur speciation and metal transport. None of this is discussed in the paper, apart from the caution given at the end of the manuscript L180-185.

At 700 °C, there is Cu-bearing fluid, but at 800–900 °C the fluid is Cu-free. It is likely a combined effect of firstly, the increase in the pyrrhotites ability to dissolve Cu (Fig. 3) and possibly, the increased H_2O activity in the fluid, which slightly reduces the H_2S activity and the ability to dissolve Cu complexes. We have discussed the Cu-distribution at the 800–900 °C experiments in the revised manuscript lines 253–262 and later in natural context in lines 324–330.

The second issue with the paper is that without detail reading of the Supplementary information the context of the main text is lost.

L86-89 Summarise well my comment above.

Although the experimental approach in understanding of ore deposits is very important the format of the paper would be more suitable for another journal which does not have restriction for the length of the main text, number of references and visuals.

The manuscript was initially submitted to Nature Geoscience, which has much smaller word limit than Nature Communications. We did not modify the manuscript to the Nature Communications format as we used the transfer portal as suggested by Rebecca Neely, the Editor of Nature Geoscience ("Nature Communications is the Nature Research flagship Open Access journal. If you would like this work to be considered for publication there, you can easily transfer the manuscript by following the instructions below. It is not necessary to reformat your paper").

In the revised manuscript, we have taken advantage of the much larger word limits of the Nature Communication and moved much of the information initially in the Supplementary Information to the main body of the manuscript.

Minor comments

Fig1 -SEM images are very poor quality (with 3 μ m scale). Reduce magnification if it is not possible to achieve better focus at the current one.

On image c, pyrrhotite is in the centre and what is the surrounding phase with irregular shape?

Unfortunately, the first author was not able to produce better quality BSE images and due to the corona virus situation the FE-SEM is not available. For this reason, we cannot get new BSE images. Still the quality is better than would be expected from a traditional thermionic filament-sourced based SEM at this magnification.

Fig2 – Is Cu in wt% or ppm?

This figure has been removed from the revised manuscript.

L118-119 is it "potentially all carbon" or as "large proportion of carbon, sulphur and copper" repetition?

Yes, there was repetition there. The interpretation about the carbon mobilization has changed in the revised version as we identified graphite among the experiment products with Raman spectroscopy. The new interpretation of the speciation is based on thermodynamic modeling of fluid composition with the PerpleX software (lines 208–243).

Compositions normalised to 100 can be misleading, especially for melts, it would be beneficial for the reader to see also data before the normalisation to estimate true Cu content of the sulphide liquid and as well as melt composition itself.

Unfortunately the FE-SEM measurements are normalized to 100 automatically by the measurement software (Oxford Instruments INCA) used at the Geological Survey of Finland.

Reviewer #3 (Remarks to the Author):

Review of Nature Communications 243616_0

General comments. I have always been resistant to the idea that metamorphic fluids could be assimilated by magmas, moving up temperature gradients and possibly also fighting pressure gradients surrounding degassing magmas that are expelling fluid with significant overpressure above ambient lithostatic pressure. Instead I've favored bulk assimilation of the whole rock instead of selective transfer of S into magmas. This article is the first to show me a convincing mechanism for the addition of S- and metal-bearing fluid to a cooling magmatic intrusion. For this reason, I think it is an important piece of work and it should be published.

However there are some problems that should be corrected before the work can be accepted. These involve:

- burial of the most interesting ideas and the over-arching story in the supplementary files. Move these front and centre and build the discussion so that it provides the scaffolding required to show why these ideas have merit
- failure to address fluid composition
- failure to address all possible contributions to the mass balance

The whole section from L137 suffers from the lack of a simple description of the mechanism proposed for transfer of metals in a S- and Cu-rich fluid from the lower parts of the thermal aureole up to a hotter zone where these metals can be added to sulfide melt and then perhaps further elevated, with the attached sulfide droplets, to the base of the intrusion where they may be assimilated. I get the picture (I think) having read the supplement first, but the ordinary reader will never see the supplement where the story is really fleshed out. This really needs to be changed so that readers can easily grasp what is being proposed. The most interesting ideas are totally buried in the supplement where only the nerdiest of the nerds will ever find them.

We have added a new section "A revised model for black shale assimilation" to the revised manuscript. In this section, we try to give a holistic picture of the assimilation model and what we envision to take place at which parts of the contact aureole as the temperature increases (lines 312–354).

There is a lot of literature out there on C-O-H-S fluids and it should be consulted to strengthen the part of the paper concerned with fluid composition. Alternatively, the authors could throw up their hands and say that the fluid composition is unknowable because there are insufficient constraints, and the article would still have documented a transfer of Cu from solids to this undefined fluid, and the rest of the story might still hold water. pardon the pun. I would prefer to see either a solid discussion of fluid composition or an admission that all that can be guessed at is the approximate Cu:S:C ratio of this fluid.

We performed additional Raman spectroscopy measurements to determine that graphite was stable phase in all the experiments, which is stated in the revised manuscript lines 124 and 436–438. These results are available in the revised version of Supplementary Information section "Identification of graphite and ferrogdrite using Raman spectroscopy" and the method is described in the manuscript (lines 534–545). We used this information to constrain the oxygen fugacity during the experiments. We also constrained the sulphur fugacity in the experiments based on the metal to sulphur ratios of the Cu-bearing pyrrhotite at the 700 °C experiment (lines 208–217). With this new information, we were able to produce a thermodynamically constrained fluid speciation model with PerpleX (lines 218–241 and Fig. 4)

The mass balance in the supplementary file only considers Cu and S distribution between sulfide phases and fluid, totally neglecting the possibility that both of these elements may enter silicate melt or that Cu may enter silicate solids (they do note in the text that Ni partitions into silicates). The mass balance looks nice but it is possibly incomplete, and it needs to be buttressed by some very forceful arguments why silicate melt and solids can be ignored, or modified to take them into account explicitly.

LA-ICP-MS was used to measure the capsule compositions and based on these measurements, there was no measurable loss of Cu or S to the capsule from the sample material in any of the experiments (see Supplementary Information sections "Analysis of Fe-S-Cu diffusion in the experiment capsule using LA-ICP-MS"). In the revised manuscript, we mention that we considered the capsules in lines 174–176. We also considered the possible effects of silicate phases to the Cu and S mass balance with the conclusion that silicate phases play no significant role in the distribution of these elements at 700 °C experiment.

In general I think we should avoid referring to minerals in plural, like "pyrrhotites" instead of calling them "pyrrhotite grains". We don't refer to "quartzes". Perhaps the editor won't agree or care. It's not important.

We have changed the spelling regarding minerals accordingly in the revised version.

Main text:

I read the main text after the supplementary file, so the comments might seem a little out of sequence. Please bear with me. I don't have time to go back and rewrite the entire review, so please just keep this sequence in mind as you read the remarks I make.

L13 sulphide phases is not a compound noun and should not be hyphenated

This has been removed.

L53 assimilation of wall rock does not necessarily require that it first melt, then mix and hybridize with the mafic magma. Wall rock can also be dissolved in the mafic magma, which is a different process and much easier to accomplish. Think of sugar in tea. The sugar is not melting. It is dissolving, at a temperature far below that at which sugar melts or caramelizes.

Anhydrite dissolution has been suggested to have taken place in Noril'sk based on experimental evidence (Iacono-Marziano et al., 2017). However, we are not familiar with cases, where silicate wall-rock would have been shown to dissolve into magma in quantities that would have caused considerable changes in the magma composition. We think that this results from the fact that dissolution is limited to the interface between the magma and the wall-rock, whereas devolatilization and melting can take place much further from this interface. For the dissolution to be effective, the wall-rock should be broken into tiny pieces with large surface area and distributed to the magma. Heating of these tiny pieces would also be very efficient, hence it's difficult to define, whether dissolution or melting would be kinetically favored.

Dissolving wall-rock to magma requires that magma is undersaturated with one or some of the wall-rock oxide components. The overall heat budget of this process is not easily defined as there are two endothermic physical processes that need to take place first: 1) breaking chemical bonds between the dissolving component, and 2) breaking chemical bonds in assimilating silicate melt network. Then this is followed by an exothermic process, when new chemical bonds form between the assimilated component and the silicate melt network. So heat is still required to start the dissolution process, but the overall heat change in the magma system can either be positive or negative depending on the relative heats required by the endothermic and exothermic steps.

It is also difficult to define how trace elements would behave during dissolution. It is likely that the isotopic composition of magma change during this process in similar way as in the case of bulk or selective silicate melt assimilation. As noted in the manuscript lines 64–66, in most parts of the Duluth Complex, bulk assimilation of more than 5 wt.% is not compatible with isotopic data, and this is generally the case with many layered intrusions.

In the manuscript, we are concentrated on assimilation of wall-rocks that are mainly composed of silicates. For this reason, we are hesitant to include discussion of dissolution in the introduction as it might cause unnecessary confusion among readers.

L76 mineralization should not be pluralized

Mineralizations has been replaced with sulphide deposits or deposits throughout the manuscript.

L94 please spell out orthopyroxene on first use of the abbreviation opx: ... orthopyroxene (opx).. : anyway, why abbreviate one mineral name but no others? I think there is enough space to write the whole mineral name. opx is jargon.

We have removed all the mineral abbreviations from the main text for clarity.

L104. Ok here is the equation I was missing in the supplementary file. Understood. Note my comments there that there might also have been significant amounts of unbound H₂O in the abundant porosity visible in the SEM image of the starting material. Please comment on this possibility, and if necessary add it to your mass balance

The starting material powder was dried in an oven at 120 °C (line 429–432) before sealing it inside the capsule. This procedure avoids any unbound H₂O entering the experiment.

L109 is there no CO in these carbonic fluids? I don't recall whether CO is stable at the CH₄-CO₂ boundary at moderate pressures. Note however that in low pressure industrial applications in settings like smelters the oxidation of reduced carbon tends to generate large amounts of metastable CO, which is a potent reductant,

even though the equilibrium species might be CO₂. I have no idea whether this could be important. I'm just putting it out as something to consider and perhaps to comment on in the text.

The new fluid model shows that the main species in the fluid are CH₄, H₂O, and H₂S depending on the amount of H₂O released from dehydrating silicates (lines 229–243 and Fig. 4).

L116. I'm not satisfied with this rather vague discussion of fluid composition. What is this fluid, exactly? Is it a carbonic fluid dominated by CH₄-CO₂ and perhaps also some CO? Is there any H₂O in it? If it is a carbonic fluid, what possible relevance does the work of Heinrich on aqueous fluids have? Do sulfide complexes as envisioned by Heinrich in saline aqueous fluids exist in carbonic fluids? Perhaps the metals are transported as carbonyl complexes in the carbonic fluid (this is how Ni is refined – we know it works). Carbonic fluids are non-polar solvents utterly unlike aqueous fluids, which have high dielectric constant and promote solvation of cations and anions. They become more non-polar at high temperatures and their miscibility with carbonic fluids increases but they are still different.

We think that the response to the previous comment also provides the answer to this comment.

The Cu-Fe-S ternary doesn't have C or O or H in it, so how can it be used to estimate the composition of a C-O-H-S fluid? The fluid-pyrrhotite tie-lines in Fig 3 are, as far as I can see, fabricated to illustrate a point based on an argument rooted in a mass balance that I have pointed out below is very poorly constrained, rather than being constrained by theory or measurement. I think the authors need to say so very clearly so readers won't mistake this bit of poetic licence for fact. It might well be true but the evidence is circumstantial. See also my concerns about mass balance – how much H₂O might there be in this fluid?

In the revised version, we have discussed the S and Cu distribution in the 700 °C experiment in much more detail for the mass balance calculation (lines 170–207 in the main text and in the section “Analysis of Fe-S-Cu diffusion in the experiment capsule using LA-ICP-MS”). We are confident now that the mass balance defines S and Cu distribution at the 700 °C experiment in good precision. As we consider the fluid assimilation only at 700 °C, that is the only temperature for which we model the COHS fluid composition (in the revised version). For the higher temperature experiments (800–1000 °C) we state, based on the mass balance lines in the Fe-Cu-S ternary, that ~all Cu must be hosted in the solid sulphides.

L141. I have to admit that at first I looked at the figure and thought the pyrrhotite was a quenched liquid. But with a little careful consideration I have to agree with the authors. However I think they need to stress this point a little more carefully. They might, if there is space, cite a pair of papers by Botcharnikov et al. that described melting experiments with pyrrhotite and sulfide melt in them where they were forced to admit in a second paper that some of their putative sulfide melt globules in the first paper were actually solid pyrrhotite that just liked to have a round smooth crystal habit. If the authors don't stress this, then some readers will think they have missed something important.

In the revised manuscript, we have clarified the point that pyrrhotite is solid with citations to the suggested literature as those papers nicely show how deceptive the purely textural interpretations may be (lines 268–269).

L156. If the authors care and have space for extra citations they might also cite LeVaillant et al., (2017: PNAS) for further discussion of the compound drop phenomenon in mafic magmas. There are also two upcoming papers in JPet and in EPSL by Yao and Mungall digging much deeper into the physics of compound drop transport in crystal-free magmas and crystal mushes but these may not be published before the present manuscript. These articles are probably not necessary to cite but the authors might find them interesting or useful.

The suggested literature was relevant and helped us to provide more clear picture of the envisioned sulphide-fluid droplet transportation process in the revised manuscript lines 299–309 and 337–354.

Supplementary file:

L54 Why is the capsule in the hot end elevated by 10° even during the run? Doesn't this run the risk of it being displaced slightly by any vibration, and then not being reliably held in the hot end? Is this a typo?

The 10° elevation is standard setting and successfully used with the apparatus in question. The hot end needs to be slightly elevated to keep the hot Ar gas in the hot end of the pressure vessel and avoid turbulent mixing of the gas, which could cause temperature fluctuations. Additionally, this allows the Ar gas in the cold end to be cool, which allows the rapid quench rate.

L62 The interpretation is reasonable. Was there no detectable Cu in the capsule? I suppose that if these runs are being discarded it doesn't matter too much. Note that if Cu was truly alloyed with AgPd then it would be present in solid solution. If it is forming an intermetallic compound with fixed stoichiometry like skaergaardite then it is not an alloy, strictly speaking, but it should be easier to detect with an SEM.

Although, we measured the 700–1000 °C capsules with LA-ICP-MS to detect impurities, we did not bother to measure the capsule used in the failed 1100 °C experiment as it is irrelevant for the main outcomes of the study.

L66. Geolabs also does a wet chemical titration for ferrous iron so that ferric/ferrous ratio can be determined (I don't think it is the metavanadate method, but it is similar). But perhaps this is not critically important. I'm reading the supplement first, before I see what was done with the results.

There is actually data for FeO (Supplementary Microsoft Excel file, below the major element oxide total), we just failed to describe the method in the original submission. The method description is now added in the revised version lines 468–472.

L113. marcasite probe standard – this must be tricky to work with. Doesn't it oxidize and blow up the polished mount? Why not pyrite? not relevant to the paper. I'm just intrigued.

We only had access to Astimex's 53 Mineral Mount MINM25-53 standards, which does not include pyrite. Personal communication with Bo Johanson, the person responsible of the FE-SEM at the Geological Survey of Finland, revealed that the composition of the old pyrite standard was not homogeneous and that the marcasite has given much more consistent results. We made 19 measurements all over the marcasite standard and got very coherent compositions throughout (Supplementary Microsoft Excel file).

L136. I don't understand. What reactions are being invoked to support the assertion that volatilization of kerogen requires inputs of H₂O? Can't oxygen be provided by reduction of ferric iron (for which you don't have any data) to produce carbonic fluid of CO-CO₂? If you don't know what was the composition of the fluid then how can you be sure that it had H₂O in it? Also, was there no H in the hydrocarbons to begin with? This bit of text is too incomplete to leave as is. Then we see that pyrite-pyrrhotite reaction has liberated half of the S in the starting material, so what is the speciation of S in the fluid? It should be possible to estimate, at least approximately, the bulk composition of this fluid and estimate speciation using one of many available fluid speciation models from the internet. There may be constraints on fO₂ from the silicate and oxide mineral assemblage and there should be a constraint on fS₂ (at least an upper bound if pyrite is completely absent since you are still below the terminal breakdown temperature for pyrite melting). Again, maybe after I read the main text I will decide that this is irrelevant. See also comments on SEM images – how much unbound pore fluid was there prior to heating, and what was its role in the melting and transport processes?

The new fluid model shows that the main species in the fluid are CH₄, H₂O, and H₂S depending on the amount of H₂O released from dehydrating silicates (lines 229–243 and Fig. 4).

L177. The discussion of Ni transport from silicates into sulfide melt or solid is apt. I'm not sure about Cu, and this also leads me to wonder if the Ni story is complete. What about the possibility of Cu transport from chalcopyrite into silicate crystals or melt? Have any of the silicate minerals or the glass been analyzed for Cu and Ni? The partition coefficients for Cu and Ni from silicate melt to sulfide solids may be rather small. If sulfide melt has not yet appeared, but silicate minerals are beginning to melt, then the Cu content of the silicate melt may not be negligible.

The nickel transport is likely minimal compared to copper as it partitions to the solid (and later in liquid) sulphides within the contact aureole, which are probably of lesser importance to the assimilation in big picture. In the revised manuscript, we briefly mention minor nickel assimilation in lines 328–329 and 343–345.

Mass balances: See comments above regarding possible distribution of Cu and Ni into phases other than fluid or sulfides. Also – why can Fe not have been incorporated into oxide or silicate minerals? Fe-rich metamorphic silicates commonly form in abundance surrounding sulfides that have lost S during metamorphic devolatilization.

In the revised version of the supplementary information, we have dedicated a section “Sulphur and copper distribution in the 700 °C experiment” to this topic and the iron conservation in solid sulphides at the 700 °C is now constrained by image analysis present in the Supplementary Information section “Estimating the mass of solid sulphides in the 700 °C experiment”.

Formation of the Cu-rich sulfide melt:

I like the idea of oxidation by H₂ loss through the capsule walls. The subsequent ideas about degassing of a S-Cu-rich fluid from the cooler end of the metamorphic thermal gradient and its ascent into the hotter zone is intriguing and worth examining. I'm not sure why this important concept is relegated to the supplementary information. I hope I will see it again in the main text.

We were initially constrained by much smaller word limit, when the manuscript was submitted to Nature Geoscience. Now the detailed description of the process can be found in the manuscript lines 273–291 and Fig. 5.

Estimation of the feasibility...

The mass of Virginia Fm over 27.5 km of strike and an unknown dip extent seems like the wrong thing to be calculating. If we assume that the lower contact of the intrusion against the Virginia Fm was essentially planar, and much larger than the snippet that is preserved today at the current erosional level, it seems to me that the valuable thing to calculate is the amount of Virginia Fm present in an arbitrary vertical column of rock, let's say 1 m square and spanning the vertical thickness of the thermal aureole. This can be compared with the amount sulfide present in a typical 1 m squared vertical column of the intrusion itself to see if the amount derived from the base of the complex is similar to the amount of S in the mineralized zone.

As can be seen from the Fig. S1, the sulphide deposits are sporadically present at the lower part of the Partridge River and South Kawishiwi Intrusion. This implies that if the assimilation took place via fluid, the fluid was channeled to certain structures of the black shale, which is to be expected in metamorphic contact aureoles (lines 317–323). If we would only consider the black shale in contact with a mineralization, then we would neglect the portion of black shale beneath non-mineralized parts of the intrusion, which must also have experienced devolatilization. For this reason, we are reluctant to calculate the mass balance according to the suggestion, although we understand that our approach may be more difficult to understand.

In the revised manuscript, we use structural observations to define the thermally affected portion of the Virginia Formation, which gives better constraints to the mass balance calculation. The new Fig. 7 helps to visualize the relation between thermally affected portion relative to the portion required for sulphur intake based on the isotopic observations of previous workers. The newly formulated mass balance calculation is now in lines 366–396.

Supplementary Figures

S1 looks fine.

S2: a. there is a lot of porosity in the starting material. How much trapped pore fluid was present prior to experiment (or prior to metamorphism) and what role would this presumably H₂O-rich fluid have played in the melting process and the putative transport of fluid and sulfide melt?

We failed to address that this was a polished thin section BSE image. The pore space is actually then likely minerals that were detached during polishing. There should not be any pore fluid as the starting material powder was kept overnight in an oven at 120 °C

b. There is only one Cpy crystal in panel a. How do you know that there was not Cpy or Py in the 700 °C run product? I expect that your search was probably careful enough but it would help to support the assertion that Cpy and Py were gone if you could explain how much surface area was searched before concluding that they were completely absent.

We checked through the run product sulphides extensively and did not find any chalcopyrite. In addition, Cu in the pyrrhotites must have derived somewhere, which means that the chalcopyrite (the only host of Cu in the starting material) must have reacted. If some of the chalcopyrites reacted, then it is highly likely that all of them did, since we did not find any crystals from the 700 °C experiment material.

Supplementary data:

I would suggest that if the analytical data for sulfides is reportable at 3 significant digits, then the normalized molar quantities and ratios should also be reported at three significant digits. Reporting mole fractions of Cu of either 0.0 or 0.1 is not very informative. Furthermore, the data in the right-most columns is labeled as (A.%), but they are definitely not percentages. They seem to have been normalized to something but it is unclear how the normalization was done since the sum of metals and sulfur is quite variable, but the amount of each individual element also varies.

We have now corrected the data in the Supplementary Microsoft Excel file so that the Cu and Ni contents are shown with three significant digits. In the original version the A.% was incorrectly not normalized to 100%, but has now been corrected in the revised version.

REVIEWER COMMENTS

Reviewer #2 (Remarks to the Author):

The authors have improved the manuscript to some extent by making some extra measurements, improving figures and adding the section on experimental and analytical techniques. However, I find it is strange that they overlooked the previous experimental work on Cu-S bearing system. I remain unconvinced that the authors have satisfactorily addressed the problem of Cu-S and Fe incorporation into the capsule material in such a way as to make mass balance calculation reliable and believable.

I am also not convinced that the authors sufficiently address the fO_2 constraints in the experiments. Presence of graphite in the run products doesn't mean that the experiments were buffered by graphite. It is not the same.

All three reviewers had the same concern about Cu and S diffusion into the Au/Pd capsule based on the personal experience and literature. Your data in supplementary table show clearly that you had a serious diffusion of Cu and S into the capsule and some amount of Fe too. How can you state that you did not have any diffusion of these elements into the capsule in the answers to reviewer's comments and in the text? You are mentioning some independent analyses of Au/Pd capsule from experiments which did not contain Cu or S, what was the concentration there, you don't report it. However, Cu content in the capsule material of ≥ 1500 ppm is not a contamination by manufacture. I would be very surprised if it was and if it is the case you should change your supplier. However, even if we assume that it is contamination during the production for Au/Pd tubing, the diffusion can go both ways into the capsule material from the experimental charge and also from the capsule material to the experimental charge. Therefore, the mass balance calculations and the results will be affected one way or another. As this study's conclusions heavily rely on the mass balance calculations I find it difficult to accept them for the reasons above. Saying that, I still think that experiments are very interesting and should be published after properly addressing Cu-S lost/gain in experimental run products. However, I am not convinced now that this manuscript is of Nature Communication scope or standard.

Reviewer #3 (Remarks to the Author):

I have reviewed the revised manuscript and carefully read the authors' response to the three reviews. I am satisfied that the important point raised by the reviewers has been addressed. I remain unconvinced that assimilation requires wholesale melting of host rock rather than dissolution (what most people mean when they talk about thermal erosion) but this is a sideshow and not really important in the present context. Although it does colour some of the viewpoints raised in the discussion of mechanisms it doesn't affect the main points of the article. As I said in my first review, to my mind the most important point of this article, which justifies publication in NCOMMS, is the idea that sulfur-rich fluids possibly attached to sulfide liquid globules will spontaneously form and rise towards the contact of an igneous intrusion, thereby permitting *selective* transfer of S and metals into the magma.

Reviewer #2 (Remarks to the Author):

The authors have improved the manuscript to some extent by making some extra measurements, improving figures and adding the section on experimental and analytical techniques. However, I find it is strange that they overlooked the previous experimental work on Cu-S bearing system.

We would be delighted to cite relevant work, but the reviewer does not indicate which experimental work we overlooked or which references we are missing.

In our literature review, we found that the issue with these kinds of experiments is that, for example, the potential for diffusion varies very much in relation to the experimental conditions and the compositions of the sample and capsule materials used in the experiments. Unfortunately, many experimental studies lacked comparable data from the sample and/or capsule materials, because they either focused on elements and/or processes that were different from our study or did not analyze the materials in as much detail as we did.

Nevertheless, we have now extensively reviewed literature related to Cu and Cu-S bearing systems to address the issue raised by the reviewer and included some of these references in the manuscript supplements. We report the summary of this review below.

The diffusion of Cu from samples to capsules have been reported by various authors (e.g., Urabe 1985, Adams and Green 2006, Zajacs et al. 2011, Fellow and Canil 2012, Wang et al. 2020). Urabe (1985) conducted experiments at 800 °C and 300 MPa with fluid-saturated (H₂O-rich, Cl-bearing fluid) silicate system, which was doped with >10000 ppm Cu. In these experiments, nearly 100% of Cu was allegedly lost to Au capsule after 48h. However, Cl is possibly one of the most important species in complexing with Cu in fluids (e.g., Zajacs et al., 2011) and Urabe (1985) does not report whether they analyzed the fluid or capsule compositions after experiment, so it could be possible that the fluid was the main host for Cu. Although the experimental conditions and durations are comparable to our experiments (700–1000 °C, 200 MPa), one notable difference is that Urabe's sample material was free of sulphur, which has high potential in stabilizing Cu within the sample.

Adams and Green (2006) state that in some of their previous experiments, sample experienced Cu-loss to Au₈₀Pd₂₀, Ag₅₀Pd₅₀, and Pt capsules. They, however, provide no information of the experimental conditions, compositional data for the capsules or samples, or references for these experiments, hence we cannot evaluate the data.

Zajacs et al. (2011) report that in a six hour experiment at 1000 °C and 150 MPa, a sample composed of andesitic glass + distilled H₂O (1:1) doped with initial 5000 ppm of Cu contained only 3 ppm Cu after the experiment. Zajacs et al. (2011) do not report the composition of their andesitic glass or explicitly state if Cu was doped as a pure metal, oxide, or e.g., CuCl, so we cannot evaluate the possible role of S or Cl in stabilizing Cu

in their experiment. As the composition of the andesitic glass is not reported for this experiment, we cannot evaluate if there were e.g., S or Cl present, which could have introduced Cu-bonding ligands in the fluid. They also do not report if they measured the composition of the capsule material before and after the experiment and if they considered the possibility of the sample fluid to contain Cu. Zajacs et al. (2011) also state that Au, AuPd, AgPd, and Pt capsules are problematic with experiments containing Cu and S bearing sample materials, but provide no data or references, which could be used to evaluate this claim. We suppose that Zajacs et al. (2011) is the “Zajacs et al. (2001)” reference, which the reviewer suggested for us during the first revision, but which we were not able to find.

Fellows and Canil (2012) report that they observed Cu alloying with Au, Pd, and Pt capsules in their experiments at 1250–1525 °C, 1 GPa and with duration of 22–55 hours. However, the method by which Cu alloying was detected is not defined and no Cu content data from capsules (before or after experiments) is available for evaluation. They also state that the most efficient Cu alloying occurred in Pt capsules with inner graphite capsule, which would indicate that the potential for Cu-loss increases with more reducing conditions. It is not clear to us whether the experiments of Fellows and Canil (2012) were fluid-saturated, but Cu-loss through inner graphite capsule to the Pt capsule seems to indicate fluid-saturation. Again, no Cu content of possible fluid is reported, so it is difficult to evaluate if the fluid could have hosted some or all of the missing Cu.

Wang et al. (2020) conducted experiments with synthetic basalt doped with a large group of trace elements and monitored elemental diffusion into Pt capsules at 1400 °C and 1 GPa. We note already here that 1400 °C is much higher temperature than in our experiments and that diffusion is strongly temperature-dependent process as shown by the Arrhenius law ($D = A * e^{-Q/RT}$, where D is diffusion rate, A is pre-exponential factor, Q is activation energy, R is the universal gas constant, and T is temperature), in which temperature is in the exponent. Wang et al. (2020) tested different fO_2 conditions from FMQ-2 to FMQ+5 and found that oxidizing conditions (Ru-RuO₂) prohibited diffusion of most elements, but Cu always diffused strongly into the capsules. Their sample was free of S, so the effect of sulphides or sulphates in stabilizing Cu cannot be evaluated. We would also like to point out that they did not measure compositions of their experiment capsules before or after experiments, but only those of sample glasses.

We would also like to point out that Jugo et al. (1999) conducted experiments with silicate rhyolite glass, pyrrhotite, chalcopyrite, and Cl-bearing aqueous fluid in Au capsules and they report no loss of Cu from the sample and to the Au capsule. They conducted the experiments at 850 °C and 100 MPa and the experiment durations were roughly between 2–20 days, which are comparable to the conditions used in our experiments.

Since there have been reports of Cu diffusion into capsule material, many researchers have decided to use AuCu capsules with known Cu content to have an exact control on

Cu activity in the capsule material (e.g., Zajacs et al. 2011, Hsu et al. 2017, Tattich and Blundy 2017, Iveson et al. 2019). In these experiments, Cu activity in capsules is much higher than in the sample material, which leads to Cu diffusing from the capsule to the sample. This is a very clever method of introducing Cu to Cu-free samples in experiments, which seek to define Cu partition coefficients, as the absolute Cu content in this purpose is not relevant. We cannot use this method, since we have chalcopyrite in the sample and we need the sample to have constant Cu content for the mass balance calculations as also noted by the reviewer.

Iveson et al. (2019) conducted LA-ICP-MS measurements to Au and $\text{Au}_{75}\text{Pd}_{25}$ capsule material, which had not been used in experiments and observed qualitatively from the raw counts per second spectra that both capsule materials contained Cu as an impurity (together with Zn, Pb, etc.). This means that manufacturing impurities seem to have been present in their capsules as well. From experiments that were conducted in these capsules, they report post-experiment Cu contents in their initially Cu-free samples: silicate glasses < 1 ppm and Cl-bearing fluid 24.7–35.7 ppm. Their experiments were conducted at 925–1050 °C, 220 MPa, and the durations were between 166–599 hours. With experiments conducted in $\text{Au}_{96}\text{Cu}_4$ capsules at 850–1000 °C, 220 MPa and with durations between 216–360 hours, the Cu contents in silicate glasses were 86.6–456 ppm and in Cl-bearing fluids 5993–33344 ppm. This indicates that differences in Cu activity between the sample material and capsule strongly affects the rate of Cu diffusion. In our experiments, both sample and $\text{Au}_{90}\text{Pd}_{10}$ capsule contain Cu, which probably is the reason, why we did not encounter Cu diffusion to either direction in amounts that would have been observable with the LA-ICP-MS, i.e., at level of >35–70 ppm.

Based on the cited literature above and as pointed out by the reviewers during the previous revision, Cu-loss to various noble metal capsules seems to be a common phenomenon in petrological experiments. It certainly occurred in our 1100 °C, 200 MPa experiment, which failed due to Cu alloying with Pd, as we already reported in the previous version of Supplementary Information. Note that in the 1100 °C experiment, capsule material was $\text{Au}_{80}\text{Pd}_{20}$, which hence comes from a different batch of raw material than the $\text{Au}_{90}\text{Pd}_{10}$ capsules used in the other experiments.

Like already mentioned in the beginning, in reviewing the existing literature, we found out that Cu-loss is often reported without detailed descriptions of the methods or data and for this reason it is difficult to evaluate the applicability of certain capsule materials in Cu-bearing experiments in general. For this reason, one just cannot know in advance whether the experiments are going to work out or not and this is why we have analyzed the capsule material so thoroughly to address this issue.

In the Supplementary Information, we now cite literature that we think is relevant and have added a short statement that Fe, Cu, and S exchange between samples and noble metal capsules have been detected previously and that for this reason we thoroughly analyzed our experiment capsules to observe whether any exchange had happened in

amounts that would affect our interpretations. Based on our detailed and high-precision analysis, it had not.

I remain unconvinced that the authors have satisfactorily addressed the problem of Cu-S and Fe incorporation into the capsule material in such a way as to make mass balance calculation reliable and believable.

We went through careful and detailed work in dissecting the capsule and measuring it from the interior wall and cross-sections with the LA-ICP-MS and also observed the changes in element signals during ablation (Fig. S6). These are now even more thoroughly reported in the Supplementary Information section “Analysis of Fe-S-Cu diffusion in the experiment capsules using the LA-ICP-MS data”. We are not aware of a previous study on natural Cu-S systems that would have performed capsule material measurements in as great detail as we have here (see the referenced work above). More details and explanations on why we think diffusion is not an issue for the main conclusions of the manuscript are given below.

We have measured the capsule material used in the experiments both in the interior wall (in contact with the sample material during experiment) and the cross-section of the capsule wall from different distances from the sample material. With our measurements we show that:

1) The Au₉₀Pd₁₀ capsule material contains Cu and S as impurities due to manufacturing as shown by the measurements from a capsule, which is from the same batch of material and was used in experiments that did not contain Cu or S in the sample. Indeed, according to the ETHZ laboratory, the used capsule materials often do contain impurities, unless special orders are given for the manufacturer to get rid of them. The Cu and S contents of these measurements from these reference capsules, as requested by the reviewer in later comments, are now shown in Fig. S7 and the data are included in the “Table S9” -tab of the supplementary Microsoft Excel file together with the other experiment capsule measurements.. Especially, Cu as an impurity is evident. The mass of the black shale sample powder within the capsule is ~20 mg and it contains 125 ppm Cu. This means that there is ~0.003 mg Cu inside the capsule before experiment. The capsule mass itself is ~100 mg and it contains ~1500 ppm Cu, i.e., ~0.15 mg in weight, which is orders of magnitude more than could derive from the sample. It is simply impossible that the sample could contribute the observed amount of Cu to the capsule as suggested by the reviewer.

2) The capsule interiors (in contact with the sample during experiment) have Cu and S contents within analytical 1 sigma error with the capsule cross-sections, which represent the original composition of the capsules. The diffusion rate for Cu at the experiments is too slow for it to be distributed homogeneously to the whole capsule (see Supplementary Discussion section “Analysis of Fe-S-Cu diffusion in the experiment capsules using the LA-ICP-MS data”), hence the cross-section measurements sample volume could not have acquired any Cu from the sample. The capsule used in the 800 °C experiment, shows elevated S contents in the interior wall, which suggests minor S

loss from the sample material. This matter is discussed in detail below and in the revised Supplementary Information. However, we already note here that the amount of S diffusion was negligible in the 800 °C experiment and also that we do not calculate mass balance for this experiment, but only for the 700 °C experiment.

3) The raw counts per second LA-ICP-MS data shows that the counts for Cu are constant relative to counts of Au and Pd with the whole depth of the ablation pit (Fig. S6). These measurements start from the surface of the capsule that was in direct contact with the sample material and the last signal comes from ~20 μm distance from the sample material. Because the Au and Pd concentrations in the capsule are known to be constant, this means that the Cu concentration in the capsule must also be constant relative to the distance to the sample within the measured volume. Copper and S diffusion to the capsule must follow the Arrhenius law, hence, if diffused, they should both be concentrated to the inner wall of the capsule, as there is not enough time for homogeneous distribution into the capsule for these elements. This is not observed outside of minor S diffusion in the 800 °C experiment. We have now marked the interior wall and 20 μm distance to the measurement spectrum in the Fig. S6a and we hope that this provides more clarity.

I am also not convinced that the authors sufficiently address the fO_2 constrains in the experiments. Presents of graphite in the run products doesn't not mean that the experiments were buffered by graphite. It is not the same.

The reviewer is correct here, we should have realized that graphite could have precipitated from the fluid after quenching the sample. That being said, the fO_2 constraints are of secondary importance to the main conclusions of the manuscript and were meant to give some constraints on the fluid speciation as asked by the third reviewer during the first revision. Nevertheless, we have now calculated the amount of carbon in the sample relative to H₂O and calculated the graphite-saturated COHS fluid compositions for all experiments and projected those on COH ternaries (Fig. S5). This analysis clearly shows that all the experiments were indeed graphite-saturated. The presented models also provide an estimated range for the absolute log fO_2 in each experiment.

All three reviewers had the same concern about Cu and S diffusion into the Au/Pd capsule based on the personal experience and literature.

This is true and the concern was legit and important. This is why we wanted to carefully study it and significantly improved this aspect in the manuscript submitted after the first revision. We would not have resubmitted the manuscript, if we would have found evidence of significant Cu, S, or Fe diffusion into the capsule. The third reviewer was happy with the resubmission and our considerations of this issue. Unfortunately, we did not receive additional comments from the 1st reviewer.

Your data in supplementary table show clearly that you had a serious diffusion of Cu and S into the capsule and some amount of Fe too. How can you state that you did not

have any diffusion of this elements into the capsule in the answers to reviewer's comments and in the text?

We have to disagree with the reviewer on most parts stated here. The reviewer does not explain how the data shows that there was diffusion of Cu and S into the capsule and it is unclear to us how our data can be interpreted in that way. As already stated above, based on our LA-ICP-MS measurements, there was no diffusion of Cu into the AuPd capsules. We have shown that the AuPd capsules contain Cu and S as impurities due to manufacturing. This is evident as measurements far from the inner wall (i.e., the cross-section measurements) also show elevated Cu and S contents. We also stated that we had measured Au₉₀Pd₁₀ capsule from the same batch of raw material with the capsules used in our experiments and these measurements show similar amounts of Cu and S. We regret only mentioning this and not providing the raw data in the previous submissions. As shown in Fig. S7, the capsule interior and cross-section measurements are within the analytical 1 sigma error. Finally, there is no indication for any variation of Cu content relative to the distance in the interior wall measurements (Fig. S6).

We already discussed the Fe diffusion to the capsules in the Supplementary Information, where we clearly stated that there was Fe diffusion from the sample into the capsule, but that the amount of Fe lost from the sample material is negligibly small. We have now rephrased this in the Supplementary Information and we provide exact values for Fe loss, which are 0.003–0.008 wt.% of total Fe in the sample.

We also noted in the Supplementary Information that the AuPd capsule of the 800 °C experiment shows elevated S contents at the immediate contact with the sample material (inner wall). The other capsules show no similar increase in the raw counts per second measurement spectra (Fig. S6). We have now added measurement data from the 900 °C and 1000 °C experiment capsules to the Fig. S6 to further prove this. The Fig. S7 already showed the same thing as the S contents of the inner walls and cross-sections are generally same within 1 sigma errors for all the other capsules. In the case of the 800 °C experiment, we also discussed that the diffusion of S to the capsule is negligible. We have now rephrased this and we state that the loss of S from the sample material at 800 °C is ~1.7 wt.% relative to the total S in the black shale sample material. This means that from the original 3690 ppm of S present in the black shale sample, ~60 ppm was lost to the capsule.

You are mentioning some independent analyses of Au/Pd capsule from experiments which did not contain Cu or S, what was the concentration there, you don't report it. However, Cu content in the capsule material of ≥1500ppm is not a contamination by manufacture. I would be very surprised if it was and if it is the case you should change your supplier.

Gold and Pd used in the capsules ultimately come from natural Au and Pd deposits. As natural materials, Au and Pd contain various impurities, e.g., Au always contains Ag and Cu. Hence, Au and Pd need to be refined to a certain level of purity. There are various refining processes, which yield different purities for Au and Pd. Although refining purifies

these materials relative to the natural compositions, it also introduces new impurities such as S to the alloy. For example, 99.995 % pure Au tends to contain more elements as impurities than 99.95 % pure Au, because the impurities are introduced in the refining processes (Kinneberg et al. 1998).

Copper (together with other metals that tend to coexist with Au, Pd, or other natural materials used to produce experiment capsules) is a common impurity in experiment capsules. Producing ultra-pure noble metal raw materials for experiment capsules is very expensive and unnecessary for most experimental purposes. Few studies have actually measured and reported compositions of their experiment capsules before the capsules have been used in experiments. We were able to find one such study and it qualitatively reports Cu as an impurity in all their four different capsule raw materials (Iveson et al. 2019).

The company that provided the Au₉₀Pd₁₀ raw material for us, also produces various Cu-noble metal alloys (see the precious metal alloy list at <https://www.pxgroup.com/de/legierungen>) and insufficient cleaning of the machinery could have introduced Cu to our material. The reported Cu content of 1500–2000 ppm in our Au₉₀Pd₁₀ capsules can either be explained by contamination during the manufacturing process or insufficient refining of the Au and/or Pd raw material used to produce the capsule alloy. There are no Cu-bearing materials that could have been in contact with the Au₉₀Pd₁₀ reference capsule that we analyzed, but still this capsule contains similar amounts of Cu as the capsules that were produced from the same batch of raw material and used in our experiments.

However, even if we assume that it is contamination during the production for Au/Pd tubing, the diffusion can go both ways into the capsule material from the experimental charge and also from the capsule material to the experimental charge. Therefore, the mass balance calculations and the results will be effected one way or another. As this study's conclusions heavily rely on the mass balance calculations I find it difficult to except them for the reasons above.

As the reviewer says and based on the studies shortly reviewed above, Cu can diffuse both from the sample to the capsule and from capsule to the sample. The diffusion direction is controlled by differences in Cu activities in the sample and the capsule, i.e., the diffusion seeks to balance the Cu activities in both systems by introducing Cu from the higher activity material to the lower activity material. The diffusion rate depends on the relative activity differences between the materials (the larger the difference, the faster the diffusion, see Iveson et al. 2019). It is difficult to know beforehand the activities of Cu in capsule material and in experimental fluids, hence capsule material should always be measured for compositional changes after experiment. We are grateful to the reviewers, who pointed this out to us in the first reviews.

In our LA-ICP-MS raw counts per second data, we found no differences in Cu/Pd or Cu/Au in the interior walls (Fig. S6). If there was Cu diffusion of the order of a few tens

of ppm to either direction, this should be clearly evident in the spectra as can be seen for S and Fe in the 800 °C experiment capsule (Fig. S6b).

Saying that, I still think that experiments are very interesting and should be published after properly addressing Cu-S lost/gain in experimental run products. However, I am not convinced now that this manuscript of Nature Communication scope or standard.

We hope that we have now properly addressed the issue with Cu, S, and Fe mass balance. This is the first study to experimentally show that both S and Cu are selectively mobilized by devolatilization fluids in black shales, which makes this fluid as a likely candidate as the main medium for S and Cu assimilation in these systems. Additionally, by attaching to the solid and liquid sulfides, the fluid also seems to aid in buoyant transportation of sulfides from the wall-rock to the magma, which has not been previously suggested elsewhere. The reviewer #3 agrees with us on this matter. Finally, we are not aware of a study on similar systems that would have analyzed the capsule materials and quantify possible sample/capsule diffusion in as much detail as we have here. We hope that the presented methods would be a highly cited benchmark for any similar studies in the future.

References:

- Adam, J. and Green, T., 2006, Trace element partitioning between mica- and amphibole-bearing garnet lherzolite and hydrous basanitic melts: 1. Experimental results and the investigation of controls on partitioning behaviour. *Contributions to Mineralogy and Petrology*, v. 152, p. 1–17, <https://doi.org/10.1007/s00410-006-0085-4>.
- Fellows, S. A. and Canil, D., 2012, Experimental study of the partitioning of Cu during partial melting of Earth's mantle. *Earth and Planetary Science Letters*, v. 337–338, p. 133–143, <http://dx.doi.org/10.1016/j.epsl.2012.05.039>.
- Hsu, Y.-J., Zajacs, Z., Ulmer, P., and Heinrich, C. A., 2017, Copper partitioning between silicate melts and amphibole: Experimental insight into magma evolution leading to porphyry copper ore formation. *Chemical Geology*, v. 448, p. 151–163, <http://dx.doi.org/10.1016/j.chemgeo.2016.11.019>.
- Iveson, A. A., Webster, J. D., Rowe, M. C., and Neill, O. K., 2019, Fluid-melt trace-element partitioning behaviour between evolved melts and aqueous fluids: Experimental constraints on the magmatic-hydrothermal transport of metals. *Chemical Geology*, v. 516, p. 18–41, <https://doi.org/10.1016/j.chemgeo.2019.03.029>.
- Jugo, P. J., Candela, P. A., and Piccoli, P. M., 1999, Magmatic sulfides and Au:Cu ratios in porphyry deposits: an experimental study of copper and gold partitioning at 850 °C, 100 MPa in a haplogranitic melt–pyrrhotite–intermediate solid solution–gold metal assemblage, at gas saturation. *Lithos*, v. 46, p. 573–589, [https://doi.org/10.1016/S0024-4937\(98\)00083-8](https://doi.org/10.1016/S0024-4937(98)00083-8)
- Kinneberg, D. J., Williams, S. R., and Agarwal, D. P., 1998, Origin and effects of impurities in high purity gold. *Gold Bulletin*, v. 31, p. 58–67, <https://doi.org/10.1007/BF03214762>.
- Tattich, B. C. and Blundy, J. D., 2017, Cu-Mo partitioning between felsic and saline-

- aqueous fluids as a function of $X_{\text{NaCl}_{\text{aq}}}$, f_{O_2} , and f_{S_2} . *American Mineralogist*, v. 102, p. 1987–2006, <https://doi.org/10.2138/am-2017-5998>.
- Urabe, T., 1985, Aluminous Granite as a Source Magma of Hydrothermal Ore Deposits: An Experimental Study. *Economic Geology*, v. 80, p. 148–157, <https://doi.org/10.2113/gsecongeo.80.1.148>.
- Wang, J., Xiong, X., Zhang, L., and Takahashi, E., 2020, Element loss to platinum capsules in high-temperature–pressure experiments. *American Mineralogist*, v. 105, p. 1593–1597, <https://doi.org/10.2138/am-2020-7580>.
- Zajacs, Z., Seo, J. H., Candela, P. A., Piccoli, P. M., and Tossell, J. A., 2011, The solubility of copper in high-temperature magmatic vapors: A quest for the significance of various chloride and sulfide complexes. *Geochimica et Cosmochimica Acta*, v. 75, p. 2811–2827, <https://doi.org/10.1016/j.gca.2011.02.029>.

Reviewer #3 (Remarks to the Author):

I have reviewed the revised manuscript and carefully read the authors' response to the three reviews. I am satisfied that the important point raised by the reviewers have been addressed.

We thank the reviewer #3 for having reviewed the revised manuscript and our previous responses to the commentary. We are pleased to see that our modifications to the previous version of the manuscript have been well received.

I remain unconvinced that assimilation requires wholesale melting of host rock rather than dissolution (what most people mean when they talk about thermal erosion) but this is a sideshow and not really important in the present context. Although it does colour some of the viewpoints raised in the discussion of mechanisms it doesn't affect the main points of the article.

We agree that the matter of melting vs. dissolution is an important topic, but a sideshow in the context of our manuscript. We think that including discussion of this topic to our manuscript would evoke unnecessary complexity without affecting the interpretation of any our experiments or their relation to the natural processes.

As I said in my first review, to my mind the most important point of this article, which justifies publication in NCOMMS, is the idea that sulfur-rich fluids possibly attached to sulfide liquid globules will spontaneously form and rise towards the contact of an igneous intrusion, thereby permitting *selective* transfer of S and metals into the magma.

We are pleased to see that the reviewer shares our view that the findings are of interest to the research community.

REVIEWERS' COMMENTS

Reviewer #3 (Remarks to the Author):

The authors have done a great job of laying out their mass balance calculations as well as could be expected, using sophisticated models of fluid composition. The issue of Cu transport to or from the capsule materials has been covered exhaustively. In my opinion, the authors have done their homework and laid out all of the arguments for their readers to see. It will be up to readers to judge for themselves whether or not to accept all of the interpretations. The authors have done everyone in the experimental world a big service by collecting all of the documented information about Cu in capsule materials and laying it out for us.

The central ideas of the manuscript are not challenged by uncertainties in the deportment of Cu during the experiments - it is clear that a lot of S and some Cu would enter a mobile fluid phase and migrate toward the basal contact of the intrusion. This is a very important qualitative result that absolutely should be published, and it certainly meets the bar in my opinion for NCOMMS.

Response to reviewers comments

Reviewer #3 (Remarks to the Author):

The authors have done a great job of laying out their mass balance calculations as well as could be expected, using sophisticated models of fluid composition. The issue of Cu transport to or from the capsule materials has been covered exhaustively. In my opinion, the authors have done their homework and laid out all of the arguments for their readers to see. It will be up to readers to judge for themselves whether or not to accept all of the interpretations. The authors have done everyone in the experimental world a big service by collecting all of the documented information about Cu in capsule materials and laying it out for us.

The central ideas of the manuscript are not challenged by uncertainties in the deportment of Cu during the experiments - it is clear that a lot of S and some Cu would enter a mobile fluid phase and migrate toward the basal contact of the intrusion. This is a very important qualitative result that absolutely should be published, and it certainly meets the bar in my opinion for NCOMMS.

We thank the reviewer for evaluating the manuscript and for acknowledging the work we did to address the comments from the previous revisions.